# Elevated Levels of Lamin A Promote HR and NHEJ-Mediated Repair Mechanisms in High-Grade Ovarian Serous Carcinoma Cell Line

**DOI:** 10.3390/cells12050757

**Published:** 2023-02-27

**Authors:** Duhita Sengupta, Asima Mukhopadhyay, Kaushik Sengupta

**Affiliations:** 1Biophysics and Structural Genomics Division, Saha Institute of Nuclear Physics, Kolkata 700064, India; 2Homi Bhabha National Institute, Training School Complex, Anushaktinagar, Mumbai 400094, India; 3Population Health Sciences Institute, Newcastle University, Newcastle upon Tyne NE2 4BN, UK

**Keywords:** lamin A, ovarian cancer, genomic instability, etoposide, chemoresistance

## Abstract

Extensive research for the last two decades has significantly contributed to understanding the roles of lamins in the maintenance of nuclear architecture and genome organization which is drastically modified in neoplasia. It must be emphasized that alteration in lamin A/C expression and distribution is a consistent event during tumorigenesis of almost all tissues of human bodies. One of the important signatures of a cancer cell is its inability to repair DNA damage which befalls several genomic events that transform the cells to be sensitive to chemotherapeutic agents. This genomic and chromosomal instability is the most common feature found in cases of high-grade ovarian serous carcinoma. Here, we report elevated levels of lamins in OVCAR3 cells (high-grade ovarian serous carcinoma cell line) in comparison to IOSE (immortalised ovarian surface epithelial cells) and, consequently, altered damage repair machinery in OVCAR3. We have analysed the changes in global gene expression as a sequel to DNA damage induced by etoposide in ovarian carcinoma where lamin A is particularly elevated in expression and reported some differentially expressed genes associated with pathways conferring cellular proliferation and chemoresistance. We hereby establish the role of elevated lamin A in neoplastic transformation in the context of high-grade ovarian serous cancer through a combination of HR and NHEJ mechanisms.

## 1. Introduction

The nuclear lamina is a nuclear peripheral meshwork composed of lamins which are type V intermediate filament proteins [1,2]. Findings over the past few years have established that lamins not only impart mechanical stability and facilitate the binding of proteins and chromatin but also serve a wide range of nuclear functions such as genome organization [3,4], maintenance of genome stability by tethering chromatin [5], chromatin regulation, DNA replication-transcription damage, and repair [6,7,8]. Mutations in the LMNA gene encoding lamin A/C or its altered expression are found to be coupled with defects in DNA replication, transcription, and repair [8,9,10,11]. These observations, together with the fact that defects in lamin A are associated with various degenerative disorders, premature aging syndromes, and cancer, validate the notion that this protein serves as the “caretaker of the genome” [12]. Lamin A/C plays a role as a complex regulatory machine in various cancers but whether its expression alteration is a cause or a consequence of a particular type and stage of cancer calls for deeper investigation.

Ovarian cancer is the 7th leading cause of cancer mortality among women and 8th leading cause of cancer diagnosis worldwide [13]. It is the leading cause of death from gynaecological malignancies after cervical and uterine cancer [14]. High-grade serous ovarian cancer (HGSOC) is the most regular histological subtype of epithelial ovarian cancer (EOC) which is characterized by functional p53 loss in 96% of cases along with a high frequency of copy number alterations (CNAs) [15]. Interestingly, most carcinoma tissues and cell lines exhibit a heterogeneous expression pattern of lamin A/C [16,17,18]. Some studies have reported that lower lamin A levels in cancer cells are associated with a higher ability to migrate; whereas, migration was impeded by introducing lamin A ectopically in the system [19]. Another group has observed that increased levels of lamin A facilitate migration in colorectal cancer cells through an elevated level of T-plastin which on the other hand, downregulates E cadherin [20]. Taken together, these findings suggest that lamin A/C expression, whether low or high, has a chance of causing metastasis and invasion, given that low levels are associated with invasion through nuclear deformability, while high levels protect against mechanical forces and, thus, resist DNA damage-induced cell cycle arrest by assisting the recruitment of DNA damage repair proteins [21,22,23,24]. In ovarian cancer tissue specimens, high-density protein microarrays revealed enhanced expression of lamin A with unaltered cellular location [17]. A proteomic investigation was recently conducted in patients with polycystic ovarian disease to identify biomarkers for ovarian cancer [25]. Calreticulin, fibrinogen, superoxide dismutase, vimentin, malate dehydrogenase, and lamin B2 were found to be overexpressed in women with both ovarian cancer and PCOD in the study [26]. Therefore, this study was another hint for predicting lamins as probable regulators of cancer progression in the ovary.

In this work, etoposide, which is a semi-synthetic derivative of podophyllotoxin extracted from *Podophyllum peltatum* or *Podophyllum emodi*, has been used as a DNA-damaging agent [27]. It works by inhibiting the action of topoisomerase II resulting in DNA strand breaks and the induction of cytotoxic and apoptotic cell death [28]. The principal chemotherapeutic effect of etoposide lies in the fact that the permanent double-strand breaks overwhelm the cells thereby initiating cell deaths (Appendix A). Thus, etoposide changes topoisomerase II from an important enzyme to a strong cellular toxin that induces breaks in the genome facilitating mutagenesis and cell death pathways [29]. However, other mechanisms, such as chromosomal aberrations, aneuploidy, and endoreplication, may, nevertheless, play a role in cancer cells’ sensitivity to etoposide [30,31]. The roles of etoposide in mitotic catastrophe, senescence, neosis, and cell survival are yet to be fully determined and are currently being debated [32]. Oral etoposide is a common candidate as a single or combination drug for second-line and/or maintenance treatment, as it does not display cross resistance to platinum or paclitaxel, which are utilized as first-line chemotherapeutic agents in the treatment of ovarian cancer [33].

Various studies in mutant LMNA and HGPS models have established fascinating cues regarding the active regulatory roles of a functional lamin A in the maintenance of genomic stability and subsequent response to radiation and chemotherapy [34,35]. Studies suggest Zmpste24-deficient mouse embryonic fibroblasts (MEFs) have higher tendencies to acquire chromosome abnormalities and are more vulnerable to DNA-damaging chemicals [36]. Upon exposure to ionizing radiation, these cells demonstrate delayed recruitment of 53BP1 to H2AX-labeled DNA repair foci, as well as delayed resolution of these foci. In addition, progeroid fibroblasts have an abnormal build-up of the xeroderma pigmentosum group A (XPA) protein, which is partly responsible for DNA repair abnormalities, as well as poor recruitment of the double-strand break (DSB) repair proteins Rad50 and Rad51 to sites of DNA damage [37]. It was also reported that DNA breakage decreases the mobility of nucleoplasmic GFP lamin A [8,38]. Furthermore, lamin A was reported to engage chromatin through phosphorylated H2AX, induced by genome damage which was elevated following irradiation [8,39]. It was also established that LMNA inactivation affects the positional stability of DNA repair foci which is recovered by the stable expression of GFP-lamin A [8,40]. The early response of cells expressing disease-causing lamin A has been studied by Parnaik and colleagues [7]. Several mutants were exposed to DNA damage induced by cisplatin or UV, which primarily caused replicational stress due to stalled replication forks in the S phase, and it was discovered that several mutants inhibited the formation of phosphorylated H2AX at DNA repair foci and the recruitment of 53BP1 to the repair sites [7]. In untreated cells, these mutations impaired emerin localization and, more importantly, mislocalized ATR kinase [7]. These data also pose an intriguing question—whether lamin A is a critical component of the DNA damage response, such that its loss causes radiation and chemosensitivity as well as an abnormal DNA damage response, or does an abnormal DNA damage response result from lamin-induced changes in the nuclear substructure [41].

To investigate this, the DNA damage repair status in a high-grade ovarian serous cancer cell line with a high endogenous lamin A (OVCAR3) level was studied compared to a normal immortalized ovarian surface epithelial cell line with a low level of endogenous lamin A (IOSE). The global gene expression was studied upon the introduction of etoposide as a DNA damaging reagent in a high endogenous lamin A background. RNA sequencing analysis reported many genes to be differentially regulated which overlapped with some previously discovered genes in studies with etoposide treatment in breast and lung adenocarcinoma cell lines. Interestingly, the list of differentially expressed genes (DEGs) in the current study also included the expression of some signature genes, directly or indirectly linked to pathways associated with cellular proliferation, apoptosis evasion, and chemoresistance. Studies in lamin A knockdown background further validated the potential of high endogenous lamin A to directly or indirectly contribute to chemoprotection in the OVCAR3 cell line.

## 2. Material and Methods

### 2.1. Cell Lines

NIH-OVCAR3 cell line was obtained from Dr. Asima Mukhopadhyay at Tata Medical Centre and maintained in ATCC-formulated RPMI 1640 supplemented with penicillin streptomycin to a final concentration of 1% and fetal bovine serum (FBS) to a final concentration of 20%. IOSE (immortalised human ovarian surface epithelial cell line) was a kind gift from Dr. Nelly Auersperg (Canadian Ovarian Tissue Bank, BC) maintained in a 1:1 ratio by volume of Media 199 and MCDB 105 supplemented with penicillin streptomycin to a final concentration of 1% and fetal bovine serum (FBS) to a final concentration of 10%. Cells were cultured using a standard protocol and are maintained at 37 °C under 5% CO_2_.

### 2.2. Treatment with Etoposide and siRNA-Mediated Knockdown of LMNA Gene

OVCAR3 cells were treated with 100 ng/mL nocodazole for 16 h and washed with PBS thrice and incubated in the fresh media for 3 h before treatment with etoposide (final concentration:1.7 µM) for 24 h. DMSO was used as the vehicle. For the knockdown experiments, cells in 35 mm culture dishes were transfected with 60 nM of siRNA complex, and 5 µL lipofectamine 2000 was prepared as per manufacturer’s instructions and incubated for 24 h per dose for up to three doses [42]. siRNA sequences for LMNA are mentioned in Appendix A).

### 2.3. Cell Lysis and Western Blot

Cells were allowed to grow to a maximum of 75–90% confluency followed by pelleting. The pellets were washed with sterile 1X PBS and stored at −80 °C until further use. For lysis, the pellets were treated with mammalian protein extraction reagent (M-PER) along with the 1X protease inhibitor cocktail. The lysates were stored at −80 °C. Protein concentrations in cell lysates were measured using the Bradford assay. Proteins were separated using 10% SDS gel electrophoresis and transferred onto a nitrocellulose membrane. Primary antibodies along with the dilutions used in this study are mentioned in the Appendix A). Antibodies were diluted in the blocking buffer and membranes were incubated overnight at 4 °C with primary antibodies. Secondary antibody dilution was 1:400 for both antimouse and antirabbit IgG conjugated to horse radish peroxidase. Membranes were incubated with secondary antibodies for 2 h at room temperature. Blots were analysed using ImageJ (ImageJ bundled with 64-bit Java 1.8.0_112) for densitometric analysis. Experiments were performed at least 6 times.

### 2.4. Immunocytochemistry and Data Analysis

Cells were grown on sterile coverslips and on reaching 70–80% confluency, they were processed for immunostaining. Cells were fixed with 4% paraformaldehyde for 15 min, permeabilized with 0.5% Triton × 100 for 5 min, blocked, washed, and incubated with primary antibody solution containing blocking agent (5% normal goat serum and primary antibody as per dilution recommended in 1× PBS) for 2 h in a humidified environment at room temperature. Following the primary antibody, cells were incubated with a secondary antibody diluted in 1× PBS in a humidified environment for 2 h in dark at room temperature. Following similar washing steps (3 min twice) with 0.05% Tween-20 and 1× PBS, cells were mounted on glass slides with Vectashield mounting medium with an antifade agent and DAPI or PI to stain the nucleus. Propidium iodide was used at a concentration of 10 μg/mL for 40 min and, then, coverslips were mounted on glass slides with a mounting medium containing an antioxidizing agent (PPD). The slides were stored in dark at 4 °C. Primary antibodies along with the dilutions used in this study are mentioned in the Appendix A). Secondary antibodies were conjugated with Alexa Fluor 488 and Alexa Fluor 546. For confocal imaging, the slides were visualized under 63× oil immersion objectives in Zeiss LSM 710 Meta, 60× (water immersion), and 100× oil DIC N2 objective/1.40 NA/1.515 RI in NIKON TiE inverted research microscope. The images were captured in resonant mode. The excitation filters used were 450/50, 525/50, and 595/50, the first dichroic mirror used was 405/488/561. The lasers used were multi line argon–krypton mixed gas laser (λ_ex_: 488 nm), a solid state laser 100 λ_ex_—405 nm, and a solid state laser (λ_ex_—561 nm). Images were processed using Ni elements analysis AR Ver 4.13 and ImageJ software (ImageJ bundled with 64-bit Java 1.8.0_112). Immunofluorescence experiments were performed at least 6 times. Around 250 nuclei were used for quantification of fluorescence intensities.

### 2.5. Comet Assay and Data Analysis

OVCAR3 cells were grown in 100 mm dishes. Cells were treated with etoposide and DMSO (vehicle) following synchronization with nocodazole. The lysis solution for comet assay was chilled at 4 °C for at least 20 min before use. LM agarose was melted in a microwave and cooled in a water bath for at least 20 min before use. 10^5^/mL cells were combined with molten LMagarose (at 37 °C) at a ratio of 1:10 (*v*/*v*) and immediately pipetted 50 µL onto two well CometSlides™ (Trevigen, Minneapolis, MN, USA). Agarose mixed cells were spread over the sample area using the side of the pipette tips. Slides were placed at 4 °C in the dark for 15–30 min. Next, the slides were immersed in lysis solution at 4 °C for 30 min. After removing the slide from lysis buffer, excess buffers were tapped and the slides were gently immersed in 50 mL of 1X TBE buffer for 5 min, two times. Neutral electrophoresis was performed at room temperature with 1X TBE at 21 V. After electrophoresis, the slides were gently removed from the tray and washed with neutralizing buffer (0.4 M Tris-HCl, pH 7.5) for 5 min followed by washing in distilled H_2_O for 5 min. Finally, the slides were stained with propidium iodide (20 µg/mL) and visualized under 20X DIC N2 objective in NIKON TiE inverted research microscope. Tail lengths and intensities were measured in a semiautomated manner using ImageJ software (ImageJ bundled with 64-bit Java 1.8.0_112.15 nuclei) from each field and a total of 500 nuclei were analysed. Ten such fields were used for quantification. Each is selected as representative independent experiment. Experiments were performed in triplicates. Ten to fifteen cells were analysed in each field. Twenty such fields were studied for each sample.

### 2.6. Replication Assay by Analysing BrdU Incorporation and Data Analysis

OVCAR3 cells were treated with 100 ng/mL nocodazole for 16 h and washed with PBS thrice and incubated in the fresh media for 3 h before treatment with etoposide for 24 h. DMSO was used as the vehicle. Ten µM of BrdU was introduced in culture media after treatment. After 2 h of BrdU treatment, cells were fixed with methanol and incubated in 2N hydrochloric acid for 30 min followed by 10 min of neutralization in 0.1 M phosphate buffer. It was then washed and stained with anti BrdU antibody (Santa Cruz) following the immunocytochemistry protocol described above. Images were grey scaled and thresholded to count the BrdU positive cells using the cell counter plugin in ImageJ (ImageJ bundled with 64-bit Java 1.8.0_112). Ten fields from each sample for all independent experiments were analysed.

### 2.7. Cell Proliferation with MTT Assay and Data Analysis

An equal number of cells were seeded in each well of 96 well plates and incubated at 37 °C for 24 h. OVCAR3 cells were treated with siRNA only, and siRNA with etoposide and DMSO, respectively. Cells without any treatment were used as control. Ten µL of MTT solution (Roche, Boston, MA, USA, 11465007001) was added to each well and the whole experiment was performed according to the manufacturer’s protocol. The absorbance was recorded at 570 nm with a reference wavelength of 630 nm by an ELISA reader. Experiments were performed in triplicates. The data was analysed in MS Excel.

### 2.8. Real-Time PCR and Data Analysis

RNA was synthesized using QIAGEN RNAeasy mini kit. Two µg of RNA was run on FA gel to analyse the purity and integrity of the sample. Five µg of RNA was reverse transcribed using a cDNA synthesis kit (Thermo Fischer Scientific, Waltham, MA, USA) with oligodt primers according to the manufacturer’s instructions. Real-time PCR with SYBR green detection was using applied biosystem 7500 real-time PCR system with fluorescence detection. Sequences of the primers used for qPCR are mentioned in the Appendix A). Experiments were conducted in three biological replicates every time for at least 6 times. Fold changes were measured from cycle threshold values using 2^−ΔΔCT^ method.

### 2.9. RNA Sequencing and Data Analysis

Transfected cells were washed twice in ice-cold PBS, trypsinized, and transferred to a 15 mL conical tube and centrifuged for 2 min at 1000 rpm. The supernatant was discarded and the pellet was stored at −80 °C until further use. RNA was isolated using a Pure Link mini RNA isolation kit (Thermo Fisher Scientific, USA). mRNA library preparation was performed using Kapa Hyperprep stranded library preparation kit (Roche, Basel, Switzerland) followed by sequencing on Novaseq 6000 (Illumina, Inc., San Diego, CA, USA) using 2X100 bp read length targeting 50 million paired-end reads. The raw sequence dataset was mapped to the reference genome ***homo sapiens* (human) [assembly GRCh38.p13]** using the HISAT2 package. The mapped sequences were then converted from SAM file format to BAM file format and sorted using the SAMTOOLS package. Following that, the FPKM value was calculated using the StringTie package. Following that, the DESeq2 package [43] was used to perform differential expression analysis between the control and mutant in replicates. The genes with adj. *p* values less than 0.05 and log2 fold change ± 1.5 from the differential expression analysis results between C (C1 and C2 denoting DMSO treated OVCAR3 cells in replicates) and D (D1 and D2 denoting etoposide treated OVCAR3 cells in replicates) were chosen for further downstream comparison analysis. GO and pathway enrichment [44] were studied using the DAVID tool. For each given gene list, pathway and process enrichment analysis was carried out with the following ontology sources: GO biological processes [44], KEGG pathway [45], reactome gene sets [46,47], TRRUST [47], PaGenBase [48], WikiPathways [49], and PANTHER [50] pathway. All genes in the genome have been used as the enrichment background. The network was visualized with Cytoscape (v3.1.2) with a “force-directed” layout and with edges bundled for clarity. MCODE algorithm was then applied to this network to identify neighbourhoods where proteins are densely connected.

### 2.10. Lamin A Interactome Analysis and Promoter-TF Binding Analysis by Bioinformatics Tools

Lamin A interactome was studied in BioGRID which is the biological general repository for interaction datasets and a database that archives genetic and protein interaction data from model organisms and humans [51]. The functional association networks were generated with the help of the GeneMANIA tool, whereupon inserting a specific gene list, it returns connections between the genes within the selected datasets and finds a dataset where the gene list under the experiment is highly connected [52]. The transcription factor identification was performed with the help of the ORegAnno database (Open REGulatory ANNOtation database) which is an open database for the curation of known regulatory elements from the literature. Annotations are gathered from users worldwide for various biological experiments and automatically cross-referenced against common databases such as PubMed, Entrez Gene, etc., with information regarding the original experimentation performed [53]. Promoter sequences for putative transcription factor binding analysis were collected from EPD (biological database and web resource of eukaryotic RNA polymerase II promoters with experimentally defined transcription start sites) and ChipBase V2 software [54,55].

### 2.11. Statistical Analysis

All experiments were performed in replicates. All the graphical diagrams have been plotted in GraphPad Prism 9.5.0 (730) software. Statistical significance was determined by multiple t tests (unpaired, using parametric test, assuming both samples from each row are from populations with the same SD) for densitometric analyses of western blot, immunofluorescence data, and fold change analyses of qPCR data. Paired *t* test (parametric) was used to analyse comet assay results. Ordinary one-way ANOVA test was performed by comparing the mean of each column by the mean of a control column in analysing the counts of BrdU positive cells. Two-way ANOVA (fitting a full model and using the Geiser Greenhouse correction) test was performed by comparing each cell mean with every other cell mean on that row in analysing the results of cell viability assay. For all the experiments, error bar indicates standard error of mean. *p* value outputs are in GP style [0.1234 (ns), 0.0332 (*), 0.0021 (**), 0.0002 (***), <0.0001 (****)]. Statistical analyses of RNA sequencing data were carried out using R language packages (version 4.0.2). Differences with *p* value < 0.05 were considered to be statistically significant. For pathway and process enrichment analysis, terms with a *p* value < 0.01, a minimum count of 3, and an enrichment factor > 1.5 (the enrichment factor is the ratio between the observed counts and the counts expected by chance) were collected and grouped into clusters based on their membership similarities. More specifically, *p* values were calculated based on the cumulative hypergeometric distribution [56] and q values were calculated using the Benjamini–Hochberg procedure to account for multiple testings [57]. Kappa scores [58] were used as the similarity metric when performing hierarchical clustering on the enriched terms where subtrees with a similarity of >0.3 were considered a cluster. The most statistically significant term within a cluster was chosen to represent the cluster.

## 3. Results

### 3.1. Increased Lamin A Expression in High-Grade Ovarian Carcinoma Cell Lines (OVCAR3)

The expression of lamins in healthy human individuals is heterogenous and very tissue specific [21]. According to the report on the differential distribution of lamins in normal human tissues, ovarian stromal cells exhibited moderate positive expression in immunostaining with lamin antibodies [59]. At the same time, the altered expression of lamins in different cancer subtypes is now evident from research worldwide [21]. The cancer cell line under this study (NIH-OVCAR3) is well characterized as a high-grade ovarian serous carcinoma cell line with a missense mutation in the TP53 gene (R248Q) [60] and an appropriate model system to study drug resistance in ovarian cancer. On the other hand, the ovarian surface epithelium (OSE) is a modified mesothelium that converts to most human ovarian carcinomas [61]. Hence, the control cell line used for comparative analysis is the immortalized ovarian surface epithelial cells (IOSE). Expression of A- and B-type lamin proteins was upregulated in OVCAR3 cells which were inferred from the western blot and immunofluorescence data (Figure 1). A similar phenomenon was also reflected in mRNA levels by qRT-PCR (Figure 1). Elevation in lamin A and lamin B levels are visibly clear both in the nuclear rim and nucleoplasm in OVCAR3 nuclei with respect to IOSE, although an abundance of nucleoplasmic lamin A is more prominent in OVCAR3 nuclei. This is even more intriguing as interaction with the transcription factors is mostly carried out by the nucleoplasmic lamin A [62] (Appendix A). However, the extent of lamin A overexpression was significant and consistent in all the experimental and biological replicates, unlike lamin B. In addition, there is a prominent elevation in the size of the OVCAR3 nuclei as observed in the immunofluorescence data which corroborates the previous study from our lab in patients diagnosed with ovarian cancer [63]. Prioritizing the extent of upregulation and statistical significance, this study further focuses on the elevated endogenous expression of lamin A in OVCAR3 cells compared to IOSE cells and its effect.

### 3.2. DNA Damage Repair Status in Normal and Ovarian Cancer Cell Lines

In the studies on laminopathies, especially on progeria that is associated with deficiency of a functional lamin A, evidence of DNA repair abnormalities first appeared along with baseline DNA damage, increased sensitivity to DNA damaging agents, chromosomal aberrations, and in some cases, a constitutively activated repair mechanism [36,64,65]. Given the function of A-type lamins in the processing of long-range NHEJ processes, such as deprotected telomeres and the stabilization of 53BP1, we anticipated that lamin A expressional changes might also modulate the overall repair apparatus [66]. Therefore, we checked the damage repair status in both cell lines. As the phosphorylation of the H2AX (γH2AX) in the surrounding chromatin is the first sign of genome damage, we checked the expression of γH2AX in both the cell lines with western blot and immunofluorescence (Figure 2). To begin with, the distribution of γH2AX was similar in both the cell lines in the absence of any DNA damaging agent. We further selected some proteins from HR (homologous recombination) and some from NHEJ (nonhomologous end joining) to check the expression of the commonly studied damage repair mediators in the cell lines. Interestingly, HR proteins (BRCA1, Rad51) exhibited elevated expression in OVCAR3 cells (Figure 2, Appendix A). The expression of Ku70 was similar in both cell lines which goes in line with another study by Langland et al., 2010 [67]. Although a recent transcriptomic study reports high NHEJ activity in OVCAR3 cells, specifically XRCC6 (Ku70) expression was found to be low in the cell line (OVCAR3) compared to the other 12 ovarian cancer cell lines carrying various mutations. It was speculated that Ku70 expression may not be the appropriate determinant of a repair mode due to the other complex genotypic and phenotypic alterations in ovarian cancers [68]. As reported by Gonzalo et al., a deficiency in A-type lamins would result in increased activity of the p130/E2F4 repressor complex and repression of the RAD51 and BRCA1 genes [66]. Our observation also followed a similar trend and we could find a higher expression of HR mediators in OVCAR3 cells with a high endogenous expression of lamin A without exposure to any form of genome damage (Figure 2, Appendix A).

### 3.3. DNA Damage in OVCAR3 and Associated Responses

Next, we aimed to study the cells’ response to chemically induced DNA damage. For this, we have used a well-studied topoisomerase inhibitor (Appendix A) etoposide as the chemical DNA damaging agent, which is also a routinely used drug in the conventional treatment of ovarian cancer [69]. The dose and duration of etoposide treatment in ovarian cancer cell lines were optimized by different groups for clinical studies in patients or in vitro [70,71,72]. The specific dose and duration for this study were 1.7 µM in DMSO for 24 h. It was selected in a way so that it brings about a significant amount of DNA damage, which was validated by comet assay and quantification of the change in γH2AX expression by immunoblot and immunofluorescence (Appendix A). As observed from the increased tail lengths of the comets in the etoposide treated OVCAR3 cells, a significant amount of genome damage was confirmed (Appendix A). DMSO alone was used as a sham for the study. Expression of γH2AX in OVCAR3 increased by 50% after treatment with etoposide which was another indication of significant DNA damage (Appendix A). We also checked the expression of PCNA which was elevated after treatment with the anticancer drug etoposide (Appendix A) which strongly affects the functional organization of S-phase nuclei, leading to the disassembly of replication factories and the redistribution of replicative factors resulting in the formation of DNA repair foci [73]. Elevation in PCNA level was maximum in 24 h and decreased slightly after 48 h (Appendix A). This experiment was performed initially to optimize the duration of etoposide treatment. All the experiments afterward have been performed with 24 h of etoposide treatment. The expressions of γH2AX and PCNA were analysed by western blot as well (Appendix A). Since OVCAR3 cells are isolated from the highly aggressive ovarian carcinoma background and are one of the desired model cells to study chemoresistance, the dose and duration of etoposide were specifically optimized to elaborately study the cells’ response to combating the damage. We also studied the expression of the common repair mediators in OVCAR3 cells after treatment with etoposide. Interestingly, both HR (BRCA1, Rad51) and NHEJ (Ku70) mediators exhibited an increase in expression following treatment with etoposide, indicating successful activation of damage-induced repair (Appendix A).

### 3.4. Analysis of Differentially Expressed Genes in Etoposide-Treated OVCAR3 Cells

High throughput next generation sequencing data from this study have been deposited in NCBI’s gene expression omnibus and are accessible through GEO series accession number GSE211529 (https://www.ncbi.nlm.nih.gov/geo/query/acc.cgi?acc=GSE211529 (accessed on 23 August 2022)). The raw sequence dataset is mapped to the reference genome using the HISAT2 package and a high mapping rate of 91–92% and 92% of reads mapped to the reference genome were obtained for DMSO and etoposide treated OVCAR3 cell samples, respectively (Appendix A). As part of the analytical pipeline, most sequencers routinely provide a QC report. In this case, FastQC generated a quality control report that could identify issues that originate in the sequencer or the starting library material. Different quality values of each sample are summarized in the table provided in Appendix A. Out of the total number of genes identified from the DeSeq2 gene count analysis (supplementary information file S2), the ones with adj. *p* values of less than 0.05 and a fold change of ±1.5 from the differential expression analysis results between the DMSO treated (control) and etoposide treated cell samples were chosen for further downstream comparison analysis. The heatmap and volcano plot for the top 205 selected genes are shown in Figure 3A,B. Interestingly, gene ontology revealed that most of the differentially expressed genes were primarily involved in DNA damage/telomere stress induced senescence, DNA double-strand break repair, cell cycle regulation, DNA packaging, DNA replication, transcriptional misregulation in cancer, telomere organization, negative regulation of DNA recombination, Rap1 signalling pathway, negative regulation of DNA metabolic processes, etc. Differentially expressed genes from various pathways were further validated by qRT-PCR (Figure 3C,D). The genes and the corresponding pathways are given below(Table 1).

Interestingly, along with the upregulated expression of genes associated with cell cycle regulation telomere maintenance DNA replication packaging, we also encountered increased expression in PI3K-Akt signalling mediators (FGF2, THBS1, TLR2), TGF-beta signalling mediators (THBS1), and genes such as BIRC3 which points to chemoresistance and apoptotic evasion [74,75,76,77,78,79,80,81].

### 3.5. Functional Enrichment and Protein-Protein Interactome Network Analysis

Genes from the heatmap (Figure 3A) were selected for further analysis. A gene ontology (GO) analysis revealed that cellular functions associated with DNA damage/telomere stress induced senescence, DNA double-strand break repair, cell cycle regulation, DNA packaging, DNA replication, transcriptional misregulation in cancer, telomere organization, negative regulation of DNA recombination, Rap1 signalling pathway, and negative regulation of DNA metabolic processes were primarily affected (Figure 4A). All genes in the genome have been used as the enrichment background. The most statistically significant term within a cluster is chosen to represent the cluster (Figure 4B). The terms within each cluster are exported to a table in supporting information file S3. To further elucidate the relationships between the terms, a subset of enriched terms has been selected and rendered as a network plot, where terms with a similarity >0.3 are connected by edges. We selected the terms with the best *p* values from each of the 20 clusters, with the constraint that there were no more than 15 terms per cluster and no more than 250 terms in total (Figure 4B).

To better understand the network, PPI enrichment analysis was performed for each given gene list using the following databases: STRING [82], BioGrid [51], OmniPath [83], and InWeb_IM [83]. Only physical interactions in STRING (physical score > 0.132) and BioGrid were used. The resultant network contains the subset of proteins that form physical interactions with at least one other member in the list. If the network contains between three and five hundred proteins, the molecular complex detection (MCODE) algorithm [84] has been applied to identify densely connected network components. The MCODE networks identified for individual gene lists have been collated and are shown in Figure 4C,D. Pathway and process enrichment analysis has been applied to each MCODE component independently, and the three best-scoring terms by *p* value have been retained as the functional description of the corresponding components, shown in the tables underneath corresponding network plots within Figure 4C,D. Here, also, we observed that the genes in the PPI network were distinctly enriched in DNA damage and telomere stress-induced senescence, meiotic recombination, HDACs, and GPCR signalling events.

### 3.6. Effect of Lamin A Knockdown in OVCAR3 Cells

The study so far generated cues on the influence of etoposide as a chemical inducer of double-strand breaks in OVCAR3 cells with a high endogenous expression of lamin A. As established from the previous results, normal epithelial cells have a significantly low amount of lamin A compared to ovarian cancer cells. Therefore, to study the specific role of lamin A in this context, an siRNA-mediated knockdown of lamin A was performed in OVCAR3 cells (Figure 5A). Simultaneously, the replication assay and cell viability assay were performed after exposure to etoposide in lamin A deficient OVCAR3 cells. It was previously reported that etoposide induces S-phase accumulation, through a p53-related pathway in the mouse foetal brain [85]. Pursuing a similar observation, a greater number of BrdU-positive cells were found in etoposide-treated OVCAR3 cells (Figure 5B). Interestingly, the number decreased in the LMNA knockdown cells and the difference in the number of BrdU-positive cells was insignificant between the etoposide-treated and untreated counterparts of LMNA-deficient OVCAR3 cells (Figure 5B). Similarly, cell viability was also hindered in the knockdown condition which further deteriorated following treatment according to the data obtained from the MTT assay (Figure 5C). Genes from the majorly affected pathways were selected to be studied further in knockdown conditions to check whether lamin A has specific roles in the modulation of their mRNA expression. Interestingly enough, most of the genes which were upregulated in etoposide-treated OVCAR3 cells were downregulated in knockdown conditions except PLK1 (Figure 5D). We further checked the expression of those genes in LMNA knockdown cells both before and after treatment with etoposide. The genes which were upregulated in etoposide-treated OVCAR3 cells exhibited decreased mRNA expression after etoposide treatment in LMNA knockdown conditions (Figure 5E) indicating potential cues of direct or indirect association with lamin A in those pathways conferring cellular proliferation, apoptosis evasion, and chemoresistance. BrdU assay, MTT assay, and qPCR experiments were performed at least three times to consider results from independent biological and experimental replicates for statistical analysis.

### 3.7. Induction of Apoptotic Evasion, Cellular Proliferation and Chemoresistance by Lamin A

To correlate how the lamin A level is connected to the regulation, lamin A interactome analysis was performed to investigate whether there is any direct or indirect association of lamin A with the mediators under study or the proteins capable of regulating their expressions. BioGrid [51] was used to investigate lamin A interactome. The excel file containing information on 1601 interactions reported to date is provided in supporting information file S4. We started with the search for the known transcription factors (Human) which belong to the interactome of lamin A. We also searched for published reports regarding the interaction of lamin A with each of the 205 genes present in the heatmap which were differentially expressed in etoposide treated OVCAR3 cells. Thirty-three such genes were found to be members of the lamin A interactome. One such gene, HES1 was a known human transcription factor which is also reported to be overexpressed in advanced ovarian serous adenocarcinoma, contributing to its stemness, metastasis, and drug resistance [86,87] (Figure 6A). As confirmed from the biochemical studies, the representative mediators (PIF1, RIF1, BRCA2, FGF2, MCM10, BIRC3, THBS1, PLK1, XRCC2, POLQ, TLR2, and BRIP1) from the RNA seq data had shown severely altered expression patterns in presence and the absence of lamin A following etoposide treatment. Therefore, these genes were primarily used for this detailed survey. Firstly, we generated a functional association network of LMNA with this set of 12 genes using GeneMANIA [52]. This association data includes protein and genetic interactions, pathways, coexpression, colocalization, and domain similarity encoded by different colours. GeneMANIA also finds other genes that are related to the set of input genes, using a very large set of functional association data. Some additional genes were found as the new members to define the pathway or the specific function completely (Figure 6B). We could find direct physical interactions of lamin A with some of the factors under study and some of them were indirectly connected. We identified 20 additional interacting proteins in this network to give rise to a complete functional association. Interestingly, we could find five of the twelve factors (BIRC3, BRCA2, POLQ, BRIP1, RIF1) under study that were direct interactors of lamin A (Figure 6C). Next, we searched for the interactors of lamin A which are reported to interact with or bind to the promoters of those 12 genes. This helped us discover 12 such lamin A interactors (TP53, MYC, TEAD4, TEAD1, CTCF, YWHAQ, NCOR2, SMAD1, RUNX1, ZBTB7A, MECP2, IGFBP5) which are reported to bind the promoters of most of the genes under study (Figure 6D). The search for the regulatory elements of the genes was performed by the ORegAnno tool and ChipBase v.2 software [53,55]. This Open Regulatory Annotation database (ORegAnno) is a resource for curated regulatory annotation. This has information about regulatory regions, transcription factor binding sites, RNA binding sites, and regulatory elements (supporting information file S5). Therefore, it was evident from this exercise that either lamin A level has a direct impact on the altered expression of some of the genes or it might have regulatory roles over the expression of the transcription factors regulating the genes under study. To address this, we selected the first four transcription factors which regulate the maximum number of the genes under study and generated a separate functional association network of LMNA with this set of four transcription factors using GeneMANIA (Figure 7A). We checked the mRNA expressions of the transcription factors and found that the levels are decreased in etoposide treated OVCAR3 cells in an LMNA knockdown background with respect to etoposide treated OVCAR3 cells with a high endogenous LA (Figure 7B). With this set of information, we could speculate the possible nodes of association of lamin A in this complex regulatory mechanism. In addition, we could find additional players in the scenario. This would require confirmatory biochemical experiments to further approve the mechanistic trajectories.

## 4. Discussion

Genomic instability is one of the fundamental signatures of carcinogenicity [88], which originates from defects in DNA damage response (DDR) pathways. DDR defects can be inflicted either by genotoxic damage induced by radiation/chemotherapy in a DDR defective background or by targeting a complimentary repair pathway leading to “synthetic lethality” [89]. In this study, we have worked on NIH-OVCAR3, one of the most extensively studied ovarian cancer cell lines which was developed in 1982 from ascites of a progressive ovarian adenocarcinoma that was resistant to cyclophosphamide, cisplatin, and doxorubicin [90]. Numerous studies [91,92,93] have indicated that the cell line is typical of HGSOC (high-grade serous ovarian carcinoma). In retrospect, NIH-OVCAR3 was incorporated in the NIH NCI-60 cell line panel study (https://dtp.cancer.gov/ (accessed on 10 September 2022)), as well as extensive genomic and drug sensitivity studies, such as the Cancer Cell Line Encyclopaedia (CCLE) study conducted by the Broad Institute and funded by Novartis (https://portals.broadinstitute.org/ccle (accessed on 10 September 2022)) and the Genomics of Drug Sensitivity in Cancer (GDSC) study organized by the UK Wellcome Sanger Institute (https://www.cancerrxgene.org/ (accessed on 10 September 2022)). Our study confirmed that OVCAR3 has significantly high endogenous lamin A levels compared to normal immortalized ovarian surface epithelial cell lines and other ovarian cancer cell lines as shown earlier [93]. It must be emphasized that the regulatory roles of lamin A in maintaining genomic stability and subsequently ionizing radiation-induced responses have been documented by many researchers, especially in mutant LMNA and HGPS models [34,35,66]. In a parallel observation, it was found that HR mediators such as RAD51, BRCA1, etc. were elevated in OVCAR3 cells both as mRNA and protein levels, which supported a previous study in the lmna-/-MEF model demonstrating a compromised HR in an lmna-/-background [66].

We analysed global gene expression in OVCAR3 following treatment with etoposide as a DNA-damaging reagent in this high endogenous lamin A background. RNA sequencing analysis revealed multiple genes to be differentially regulated that overlapped with the previously reported genes in etoposide-directed therapy in breast and lung cancer cell lines [32,94]. However, no change in the gene expression of LMNA, LMNB1, and LMNB2 was observed. Interestingly, along with genes associated with double-strand break repair and telomere maintenance (BRCA2, BRIP1, RIF1, XRCC2, POLQ), the current study found some differentially expressed genes which are either directly or indirectly associated with cellular proliferation (FGF2), positive regulation of cell cycle (BRIP1, BRCA2, MCM10), apoptosis evasion (BIRC3), and chemoresistance genes linked to PI3-Akt signalling and TGFβ signalling pathways (THBS1, TLR2, BIRC3, FGF2). We have selected 12 such genes (PIF1, RIF1, BRCA2, FGF2, MCM10, BIRC3, THBS1, PLK1, XRCC2, POLQ, TLR2, and BRIP1) from the majorly affected pathways for validation. All the genes except PIF1 and PLK1 were upregulated in etoposide treated OVCAR3 cells. Further to exploring the specific role of lamin A, lamin A was knocked down in OVCAR3 cells and subsequent qRT-PCR analysis of these genes reversed the trend, except for PLK1. We further performed qRT-PCR experiments with this set of genes in a lamin A knockdown background before and after etoposide treatment which exhibited similarly decreased mRNA expressions for all the genes even after etoposide treatment. Therefore, we can conclude that an elevated expression of lamin A might lead to increased chemoresistance and aggressive metastasis of OVCAR3 directly or indirectly.

However, the complex circuitry still eluded us and we searched for ways through which lamin A may affect the expression of these factors. It is well established that lamins can affect gene expression in many ways such as controlling chromatin shape, organization at the nuclear periphery, and regulating transcriptional activity [95]. Lamins also serve as a scaffold for transcription factors such as RNA polymerase II [96,97]. Simultaneously, studies also suggest that the importance of promoter-lamina contacts in gene suppression is emphasized by the frequent exclusion of transcriptionally active gene promoters from nuclear lamina association [98]. We set sail to search for factors with reported association (direct and indirect) with lamin A via all possible modes of interaction (physical/chemical/high throughput/genetic) using bioinformatic tools to investigate the following routes:(a)Whether some of the factors have direct interaction with lamin A;(b)Whether any transcription factors of the differentially regulated genes are interacting partners of lamin A.

We explored five among the twelve factors (BIRC3, BRCA2, POLQ, BRIP1, RIF1) under study to be direct interactors of lamin A. We could also find 12 lamin A interactors (TP53, MYC, TEAD4, TEAD1, CTCF, YWHAQ, NCOR2, SMAD1, RUNX1, ZBTB7A, MECP2, IGFBP5) which are reported to bind the promoters of most of the genes under study. We further found decreased expression of some of the transcription factors in etoposide treated OVCAR3 cells under LMNA knockdown background. Therefore, we could identify the potential nodes of connection of lamin A in this intricate regulatory system.

The gene set under study was comprised of genes that have direct connections to cellular proliferation, apoptotic evasion, and chemoresistance. Most of these genes were upregulated in etoposide treated OVCAR3. Interestingly, their expressions reversed with a decrease in lamin A level. However, the lamin A level had been associated with anti-apoptosis, proliferation, and resistance to chemotherapy in various independent studies pertaining to different cancer models and therapies. Earlier studies suggest that lamins may prevent or postpone apoptosis in some cancers depending on their quantity and accessibility to caspases [99]. A dense lamina can stop or at least postpone the onset of apoptosis and prevents chromatin condensation and fragmentation. For instance, a study by Rao et al. showed that lamin A might postpone the initiation of apoptosis for 12 h by making the VEVD/VEID lamin cleavage site uncleavable by caspases. Therefore, the indirect strategy for evasion of apoptosis might be the reduction in the functional caspases [99] and/or an elevation in the level of antiapoptotic protein such as BIRC3, which is further strengthened in this current study in ovarian cancer background. Another study in PTEN-positive DU145 and the PTEN-negative LNCaP and PC3 prostate cancer cells exhibited that the protein levels of the PI3K subunits (p110 and p85), phosphor-AKT, and PTEN are directly proportional with lamin A levels. These indicate a possible role for lamin A/C proteins in prostate malignancy via the PI3K/AKT/PTEN pathway [100]. Researchers also suggest lamin A/C knockdown results are in a pause in cellular proliferation as well as a decrease in p130 levels in the nucleus. The proliferation deficit induced by lamin A/C depletion was not reversed by shRNA against p130 [101]. Studies have also shown resistance to paclitaxel-induced nuclear breakage in lamin A/C over-expressing cancer cells. It was found that decreased nuclear lamin A/C protein levels correlate with nuclear shape deformation and are a critical predictor of cancer cells’ susceptibility to paclitaxel [102]. We have encountered different players of similar processes being significantly affected by the abundance or deficiency of lamin A. We were also able to speculate possible nodes in this complex regulatory system and identified additional participants in this network. BRCA1/2 are among the genes that are essential for homologous recombination repair. BRCA2 operates almost solely in homologous recombination, in contrast to BRCA1, which has multiple activities [103,104]. Studies have found that ovarian cancer tissues have higher transcriptome-level BRCA1/2 expression which is in line with our finding [105]. The faster rate of proliferation in malignant tissues, along with genetic instability, needs a greater requirement for DNA damage repair which may be the root cause of these changes [105]. Accordingly, high-grade cancers also showed greater BRCA1/2 expression in earlier studies [105]. Gudas et al. provide credence to this idea by proposing that the proliferation of breast cancer cells is indirectly responsible for the elevation of BRCA1 expression by steroid hormones [105]. It is to be noted that OVCAR3 cells did not have any mutations in known HR repair genes, but deletions were reported in several HRR-related genes including BRCA2. However, in BRCA2 wild-type cases of ovarian cancer, a high level of BRCA2 mRNA expression is one of the determinants of chemoresistance [68,105]. Similarly, studies have demonstrated that NHEJ defects, which are independent of HR function and linked to resistance to PARP inhibitors in ex vivo primary cultures, are present in 40% of ovarian malignancies [106]. The DNA-dependent protein kinase (DNA-PK) containing Ku70, Ku80, and DNA-PK catalytic subunit (DNA-PKcs), as well as the heteromultimeric XRCC4/Ligase IV, form the base of the nonhomologous end-joining machinery [107]. Researchers have performed CGH profiling on the panel of ovarian cell lines to see if gross amplifications or deletions in these locations associated with relative X-ray sensitivity exist, because Ku70 and DNA-PK exist on chromosomes that are frequently found to be abnormal in ovarian malignancies [67]. However, no significant correlations were observed. Interestingly, in accordance with our finding, this study also could find no difference in Ku70 protein expression between OVCAR3 and IOSE cells [67]. Although, there were changes in the copy number of DNA-PKs, Ku70, and Ku80 which did not correlate with radiosensitivity [67].

Finally, the network of the interaction of lamin A has been further analysed using the miRWalk database and tools. Simultaneously, we searched for miRNAs that act as regulatory elements of the other genes as well; hsa-miR-30a-3p was one such miRNA that has putative binding affinities for LMNA 5′UTR. Not only that, but it also has tendencies to bind CDS of BRCA2, THBS1, MCM10, POLQ, BRIP1, 3′UTR of XRCC2, and 5′UTR of TLR2. Interestingly, this miRNA was also found as a regulatory element for genes such as PTEN, SMAD4, ESR1, NRG1, etc., in the miRWalk database for ovarian carcinoma. Although this was intriguing to study, it complicates the scenario with an entirely new approach. However, it opens up avenues for future studies to look into the molecular details behind the regulation.

In summary, this study has provided for the first time the potential molecular cues (Figure 8) behind the role of lamin A in the maintenance of genomic stability, cellular viability, apoptotic evasion, and, more interestingly, resistance to DNA damaging agents or chemotherapeutic substances in the context of high-grade ovarian serous carcinoma.

## Figures and Tables

**Figure 1 cells-12-00757-f001:**
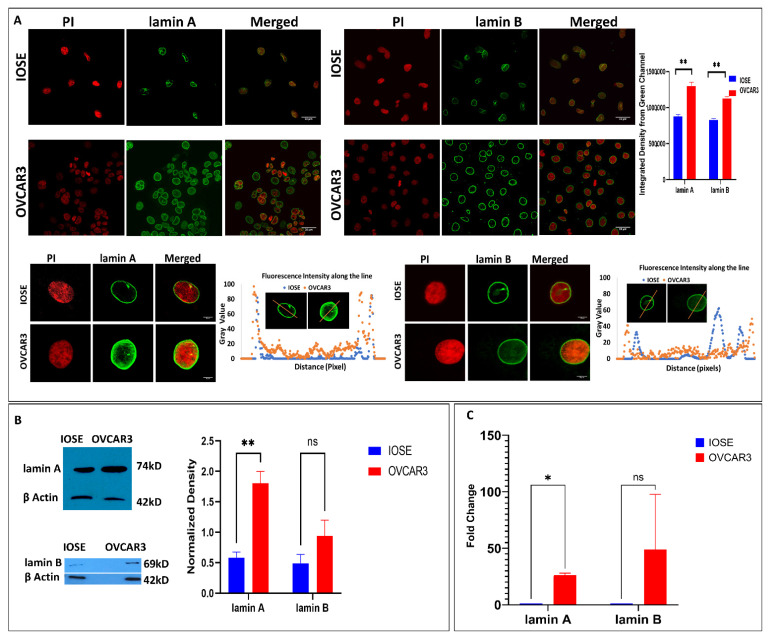
**lamin A is upregulated in OVCAR3 cells.** (**A**). Immunofluorescence images of OVCAR3 and IOSE nuclei stained with lamin A and lamin B magnification: 40×. Scale Bar: 33 µm. Bar diagram shows the integrated density from the green channels (lamin A and lamin B stained IOSE and OVCAR3 cell nuclei) of the confocal images. Error bar indicates standard error of mean. Statistical significance has been analysed by multiple *t* tests (unpaired, using parametric test, assuming both samples from each row are from populations with the same SD). *p* value output is in GP style [0.1234 (ns), 0.0332 (*), 0.0021 (**), 0.0002 (***), <0.0001 (****)]. One nucleus from each is selected as representative for the analysis of the fluorescence intensity of the green channel along the line marked across each nucleus. The first panel shows propidium iodide staining in red and the third panel shows the merged images. Magnification: 60×. Scale Bar: 5 µm. (**B**). Western blots showing the expression of lamin A and lamin B protein in OVCAR3 and IOSE cells. βActin is used as the loading control. The bar diagram depicts the normalized expression normalized to the loading control. Error bar indicates standard error of mean. Statistical significance has been analysed by multiple *t* tests (unpaired, using parametric test, assuming both samples from each row are from populations with the same SD). *p* value output is in GP style [0.1234 (ns), 0.0332 (*), 0.0021 (**), 0.0002 (***), <0.0001 (****)] (**C**). Expression of lamin A and lamin B mRNA in IOSE and OVCAR3 cells by qRT-PCR. Error bar indicates standard error of mean. Statistical significance has been analysed by multiple t tests (unpaired, using parametric test, assuming both samples from each row are from populations with the same SD). *p* value output is in GP style [0.1234 (ns), 0.0332 (*), 0.0021 (**), 0.0002 (***), <0.0001 (****)].

**Figure 2 cells-12-00757-f002:**
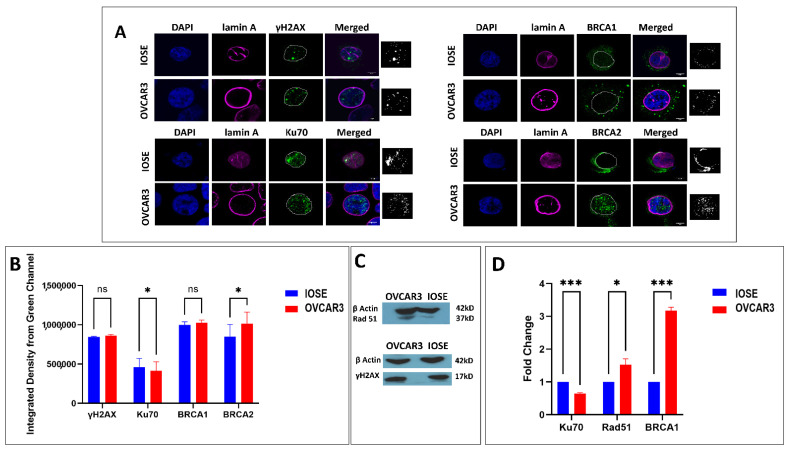
**OVCAR3 has elevated levels of HR proteins.** (**A**). Confocal micrographs showing the distribution of γH2AX and DNA damage repair proteins in IOSE and OVCAR3 cells. Magnification: 60×. Scale Bar; 5 µm. The sixth panel shows the B&W thresholding of green channel images to illuminate the count of puncta. (**B**). Integrated density from the green channel (γH2AX, Ku70, BRCA1, and BRCA2, respectively) of the confocal images. Error bar indicates standard error of mean. Statistical significance has been analysed by multiple *t* tests (unpaired, using parametric test, assuming both samples from each row are from populations with the same SD). *p* value output is in GP style [0.1234 (ns), 0.0332 (*), 0.0021 (**), 0.0002 (***), <0.0001 (****)]. (**C**). Western blots showing the level of Rad51 and γH2AX. β Actin is used as the loading control. (**D**). qPCR showing fold changes of DNA damage repair proteins in OVCAR3 cells compared to IOSE cells. Error bar indicates standard error of mean. Statistical significance has been analysed by multiple *t* tests (unpaired, using parametric test, assuming both samples from each row are from populations with the same SD). *p* value output is in GP style [0.1234 (ns), 0.0332 (*), 0.0021 (**), 0.0002 (***), <0.0001 (****)].

**Figure 3 cells-12-00757-f003:**
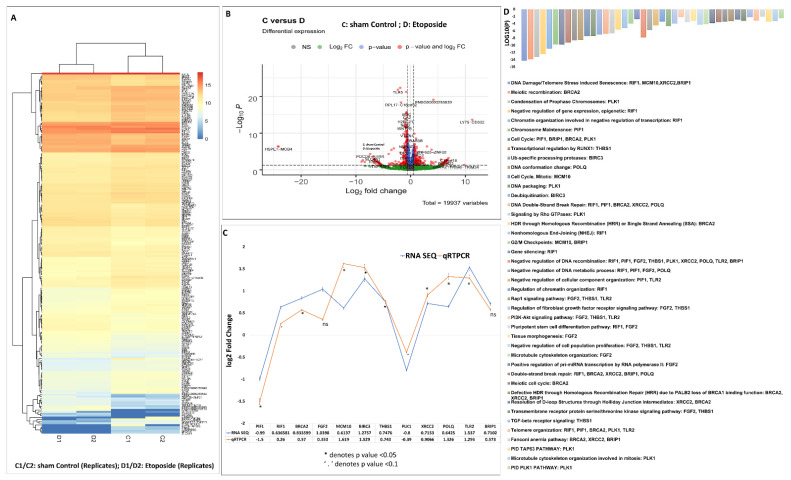
Analysis of differentially expressed genes in etoposide treated OVCAR3 cells using RNA sequencing and validation of RNA sequencing data by qRT-PCR: (**A**). The heatmap for the top 205 selected genes. C (C1 and C2 denoting DMSO (sham control) treated OVCAR3 cells replicates) vs. D (D1 and D2 denoting etoposide treated OVCAR3 cells in replicates) (**B**). Volcano plot for human samples. Volcano plot of all genes significantly regulated between the group C vs. D (log2-fold change threshold = 0.58 (FC = 1.5), Benjamini–Hochberg corrected *p* value threshold = 0.05). (**C**). Some differentially expressed genes from the majorly affected pathways (PIF1, RIF1, BRCA2, FGF2, MCM10, BIRC3, THBS1, PLK1, XRCC2, POLQ, TLR2, and BRIP1) are further validated by qRTPCR. Error bar indicates percentage error. * indicates *p* value < 0.05 and ‘.’ indicates *p* value < 0.1. (**D**). Corresponding affected pathways have been represented as a bar diagram. “Log_10_(P)” is the *p* value in log base 10.

**Figure 4 cells-12-00757-f004:**
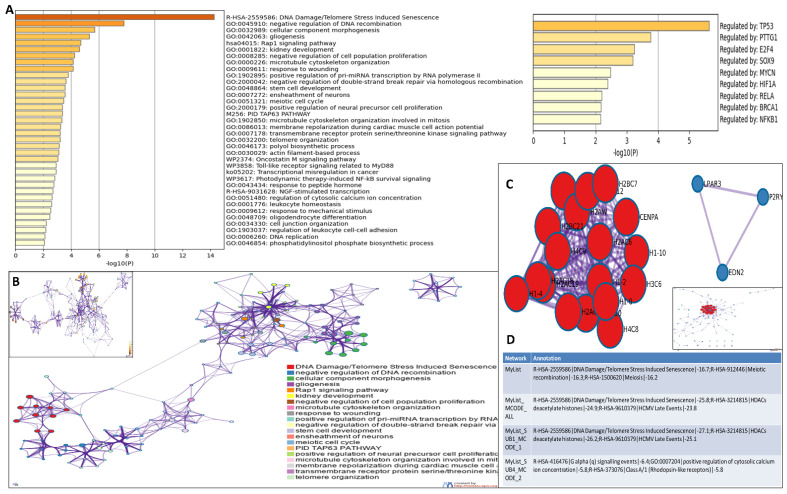
**Functional enrichment and protein-protein interactome network analysis:** (**A**). We first identified all statistically enriched terms, accumulative hypergeometric *p* values, and enrichment factors that were calculated and used for filtering. The remaining significant terms were then hierarchically clustered into a tree based on kappa statistical similarities among their gene memberships. Then 0.3 kappa score was applied as the threshold to cast the tree into term clusters. We then selected a subset of representative terms from this cluster and convert them into a network layout. More specifically, each term is represented by a circle node, where its size is proportional to the number of input genes that fall into that term, and its colour represents its cluster identity (i.e., nodes of the same colour belong to the same cluster). Terms with a similarity score > 0.3 are linked by an edge (the thickness of the edge represents the similarity score). The network is visualized with Cytoscape (v3.1.2) with a “force-directed” layout and with edge bundled for clarity. One term from each cluster is selected to have its term description shown as the label. The network of enriched terms is coloured by cluster ID, where nodes that share the same cluster ID are typically close to each other; inset shows the network of enriched terms coloured by *p* value, where terms containing more genes tend to have a more significant *p* value. (**B**). Top 21 clusters with their representative enriched terms (one per cluster). “Count” is the number of genes with membership in the given ontology term. “%” is the percentage of all of the genes that are found in the given ontology term (only input genes with at least one ontology term annotation are included in the calculation). “Log_10_(P)” is the *p* value in log base 10. “Log_10_(q)” is the multitest adjusted *p* value in log base 10. (**C**). MCODE algorithm was then applied to this network to identify neighbourhoods where proteins are densely connected. Each MCODE network is assigned a unique colour. (**D**). GO enrichment analysis was applied to each MCODE network to assign “meanings” to the network component. Protein–protein interaction network and MCODE components were identified in the gene lists.

**Figure 5 cells-12-00757-f005:**
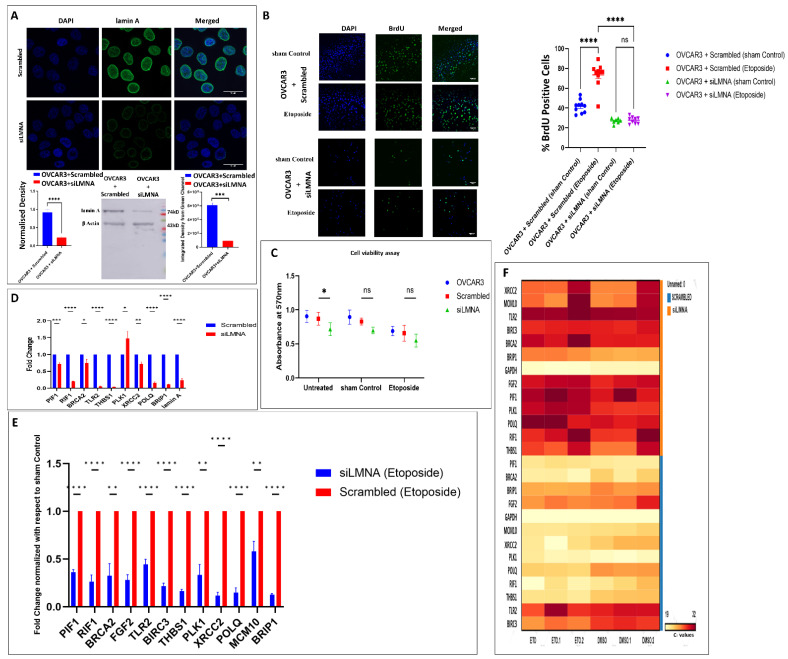
**Lamin A was knocked down in OVCAR3 cells.** (**A**). Confocal micrograph showing siRNA mediated knock down of lamin A in OVCAR3 cells. Magnification: 60×, scale bar: 10 µm. Bar diagram shows the integrated density from the green channel (lamin A stained scrambled and siLMNA treated OVCAR3 cell nuclei) of the confocal images. Error bar indicates standard error of mean. Statistical significance has been analysed by paired *t* test (parametric). *p* value output is in GP style [0.1234 (ns), 0.0332 (*), 0.0021 (**), 0.0002 (***), <0.0001 (****)]. Western blot showing siRNA mediated knock down of lamin A in OVCAR3 cells. The bar diagram depicts the quantified level of lamin A in scrambled and siLMNA transfected OVCAR3 cells. Error bar indicates standard error of mean. Statistical significance has been analysed by paired *t* test (parametric). *p* value output is in GP style [0.1234 (ns), 0.0332 (*), 0.0021 (**), 0.0002 (***), <0.0001 (****)] (**B**). Confocal micrograph showing BrdU incorporation in etoposide treated and untreated OVCAR3 cells (scrambled) and LMNA knockdown OVCAR3 cells. Magnification: 20×. Scale bar: 10 µm. The box plot depicts the percentage of BrdU-positive cells in each condition. The number of BrdU-positive cells was counted in ImageJ. Ten fields from each condition were used for quantification from each independent experiment. Error bar indicates standard error of mean. Ordinary one-way ANOVA test was performed by comparing the mean of each column by the mean of a control column. *p* value output is in GP style [0.1234 (ns), 0.0332 (*), 0.0021 (**), 0.0002 (***), <0.0001 (****)] (**C**). MTT assay showing the extent of cell viability in OVCAR3 (treated and untreated), scrambled (treated and untreated), LMNA knockdown (treated and untreated) cells. Data are plotted from three independent experiments. Error bars indicate standard error of mean. Two-way ANOVA (fitting a full model and using the Geiser Greenhouse correction) test was performed by comparing each cell mean with every other cell mean on that row. *p* value output is in GP style [0.1234 (ns), 0.0332 (*), 0.0021 (**), 0.0002 (***), <0.0001 (****)] **(D**). mRNA expressions of the representative genes selected from each of the majorly affected pathways were checked in LMNA knockdown conditions by qPCR. Error bar indicates standard error of mean. Statistical significance has been analysed by multiple *t* tests (unpaired, using parametric test, assuming both samples from each row are from populations with the same SD). *p* value output is in GP style [0.1234 (ns), 0.0332 (*), 0.0021 (**), 0.0002 (***), <0.0001 (****)] (**E**). mRNA expression of the representative genes selected from each of the majorly affected pathways were checked in LMNA knocked-down cells treated with etoposide by qPCR. Error bar indicates standard error of mean. Statistical significance has been analysed by multiple *t* tests (unpaired, using parametric test, assuming both samples from each row are from populations with the same SD). *p* value output is in GP style [0.1234 (ns), 0.0332 (*), 0.0021 (**), 0.0002 (***), <0.0001 (****)] (**F**). The heatmap for the 12 selected gene expressions based on Ct values correlated with lamin A level.

**Figure 6 cells-12-00757-f006:**
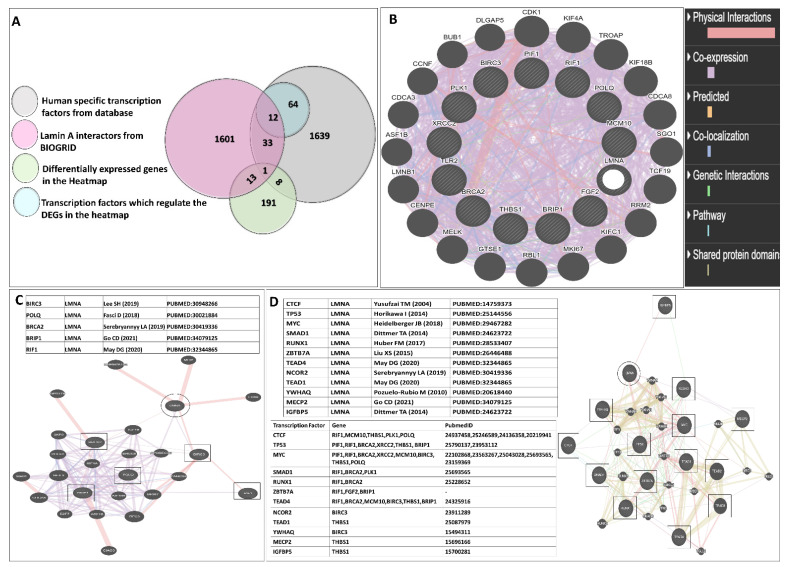
**Lamin A interactome analysis.** (**A**). Venn diagram representing functional relationships of lamin A with the differentially expressed genes from the heatmap as regulatory elements. (**B**). Functional association network of LMNA with the set of 12 representative genes from the majorly affected pathways generated with the help of the GeneMANIA tool. (**C**) Functional association network of LMNA with five of the twelve factors (BIRC3, BRCA2, POLQ, BRIP1, RIF1) under study which are direct interactors of lamin A (as investigated by BioGRID interactome analysis). (**D**) Functional association network of LMNA with twelve of its interactors (TP53, MYC, TEAD4, TEAD1, CTCF, YWHAQ, NCOR2, SMAD1, RUNX1, ZBTB7A, MECP2, IGFBP5) which are reported regulatory elements of three of the twelve genes under study.

**Figure 7 cells-12-00757-f007:**
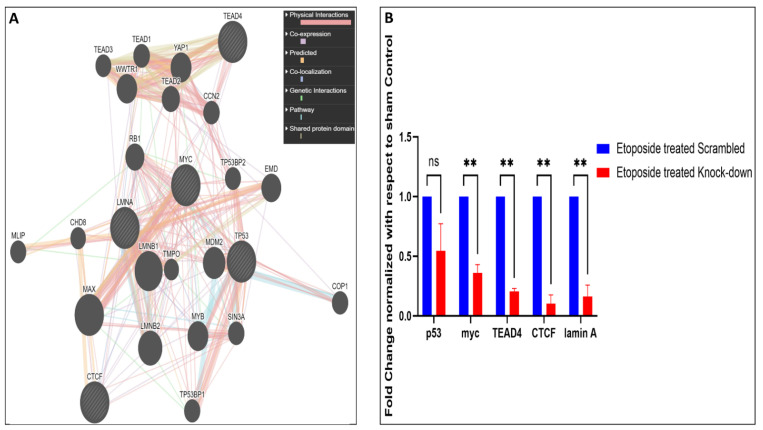
**Level of lamin A influences the expression of the genes under study:** (**A**). Functional association network of LMNA with four of its interactors (TP53, MYC, TEAD4, CTCF) which regulate most of the genes under study generated with the help of the GeneMANIA tool. (**B**). mRNA expressions of the lamin A interactors which are major regulatory elements of the genes under study were checked by qPCR in LMNA knocked-down cells treated with etoposide. Error bar indicates standard error of mean. Statistical significance has been analysed by multiple *t* tests (unpaired, using parametric test, assuming both samples from each row are from populations with the same SD). *p* value output is in GP style [0.1234 (ns), 0.0332 (*), 0.0021 (**), 0.0002 (***), <0.0001 (****)].

**Figure 8 cells-12-00757-f008:**
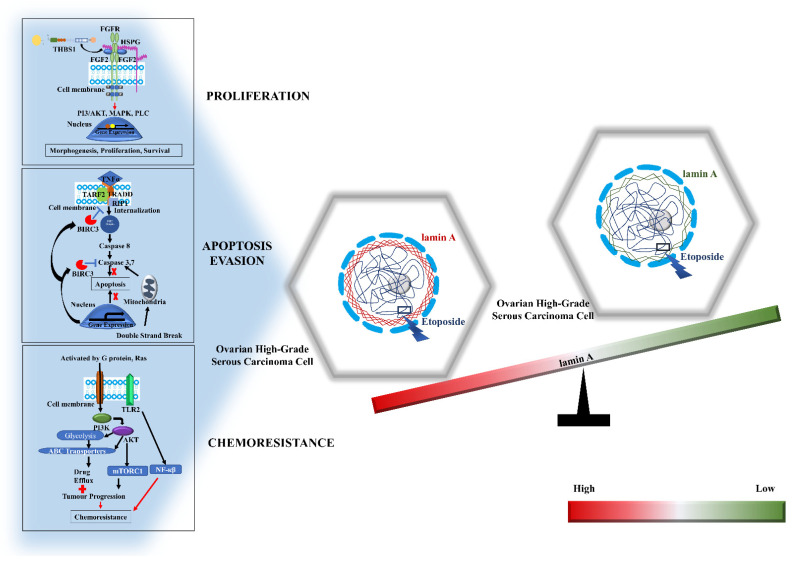
The molecules associated with the role of lamin A in the maintenance of genomic stability, cellular viability, apoptotic evasion, and, more interestingly, resistance to DNA damaging agents or chemotherapeutic substances in the context of high-grade ovarian serous carcinoma.

**Table 1 cells-12-00757-t001:** The differentially expressed genes and the associated pathways.

Gene Names	Associated Pathways
RIF1, MCM10, XRCC2, BRIP1	DNA Damage/Telomere Stress Induced Senescence
BRCA2	Meiotic recombination
PLK1	Condensation of Prophase Chromosomes, DNA packaging, Signaling by Rho GTPases, PID TAP63 pathway, Microtubule cytoskeleton organization involved in mitosis, PID PLK1 pathway
RIF1	Negative regulation of gene expression [epigenetic], Chromatin organization involved in negative regulation of transcription, Nonhomologous End-Joining (NHEJ), Gene silencing, Regulation of chromatin organization
PIF1	Chromosome Maintenance
PIF1, BRIP1, BRCA2, PLK1	Cell Cycle
THBS1	Transcriptional regulation by RUNX1, TGF-beta receptor signaling
BIRC3	Ub-specific processing proteases, Deubiquitination
POLQ	DNA conformation change
MCM10	Cell Cycle [Mitotic]
RIF1, PIF1, BRCA2, XRCC2, POLQ	DNA Double-Strand Break Repair
BRCA2	HDR through Homologous Recombination (HRR) or Single Strand Annealing (SSA), Meiotic cell cycle
MCM10, BRIP1	G2/M Checkpoints
RIF1, PIF1, FGF2, THBS1, PLK1, XRCC2, POLQ, TLR2, BRIP1	Negative regulation of DNA recombination
RIF1, PIF1, FGF2, POLQ	Negative regulation of DNA metabolic process
PIF1, TLR2	Negative regulation of cellular component organization
FGF2, THBS1, TLR2	Rap1 signaling pathway
FGF2, THBS1	Regulation of fibroblast growth factor receptor signaling pathway
FGF2, THBS1, TLR2	PI3K-Akt signaling pathway
RIF1, FGF2	Pluripotent stem cell differentiation pathway
FGF2	Tissue morphogenesis, Microtubule cytoskeleton organization, Positive regulation of pri-miRNA transcription by RNA polymerase II
FGF2, THBS1, TLR2	Negative regulation of cell population proliferation
RIF1, BRCA2, XRCC2, BRIP1, POLQ	Double-strand break repair
BRCA2, XRCC2, BRIP1	Defective HDR through Homologous Recombination Repair (HRR) due to PALB2 loss of BRCA1 binding function
XRCC2, BRCA2	Resolution of D-loop Structures through Holliday Junction Intermediates
FGF2, THBS1	Transmembrane receptor protein serine/threonine kinase signaling pathway
RIF1, PIF1, BRCA2, PLK1, TLR2	Telomere organization
BRCA2, XRCC2, BRIP1	Fanconi anemia pathway

## Data Availability

RNA Sequencing data from this study have been deposited in NCBI’s Gene Expression Omnibus and are accessible through GEO Series accession number GSE211529 (https://www.ncbi.nlm.nih.gov/geo/query/acc.cgi?acc=GSE211529 (accessed on 23 August 2022)) Rest of the data will be available upon request.

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
