# Peer review of "Elevated Levels of Lamin A Promote HR and NHEJ-Mediated Repair Mechanisms in High-Grade Ovarian Serous Carcinoma Cell Line"

_cells, 2023, doi:10.3390/cells12050757_

Round 1

Reviewer 1 Report (Previous Reviewer 2)

All the concerns have been addressed by the authors.

Author Response

All the concerns have been addressed by the authors.

Response: We heartily thank the reviewer for the positive review.

Reviewer 2 Report (Previous Reviewer 3)

The authors have edited the manuscript, addressing all comments and thus much improved the presentation of this research. While there are several open questions, especially working on molecular interactions to verify the soundness of the scientific study- this could be followed up in later work. The current study seems adequate to be a stand alone publication, as the claims have been edited to better agree with the results. The manuscript figures require format editing to make the font of graphs etc. more legible. 

Author Response

The authors have edited the manuscript, addressing all comments and thus much improved the presentation of this research. While there are several open questions, especially working on molecular interactions to verify the soundness of the scientific study- this could be followed up in later work. The current study seems adequate to be a stand alone publication, as the claims have been edited to better agree with the results. The manuscript figures require format editing to make the font of graphs etc. more legible. 

Response: We heartily thank the reviewer for the positive comment. We have edited the figures accordingly and the axes of the graphs are more legible now, we have increased the font sizes of the graph axes labels and figure labels in Figure 1A quantification graph and images, Figure 3C graph, Figure 3D graph, Figure 5A graph, Figure 5B graph, Supplementary Figure 3D graph etc in the revised version.

This manuscript is a resubmission of an earlier submission. The following is a list of the peer review reports and author responses from that submission.

Round 1

Reviewer 1 Report

Sengupta et al. show that elevated lamins levels in ovarian carcinoma cells compared to immortalized ovarian epithelial cells is associated with alterations in DNA repair machinery, such as elevated DNA repair factors -RAD51, BRCA1, and BRCA2. No elevated basal DNA damage (gH2AX) was observed in tumor or immortalized ovarian cells.

Upon Etoposide (inhibitor of topoisomerase II) treatment, the authors find that high lamins-expressing cells show higher levels of HR mediators such as BRCA1, RAD51, or Ku70. An RNAseq analysis identifies that DEGs include DNA damage/telomere stress-induced senescence, DNA DSB repair, cell cycle regulation, DNA packaging, replication, transcription, recombination, etc. Using network analysis software, they find interactions enriched in DNA damage, telomere stress-induced senescence, meiotic recombination, HDACs and GPCR signaling events.

Finally, used siRNA-mediated KD of lamin A and monitor response of cells to DNA damage (etoposide). Find less BrdU labeled cells in lamin A-depleted cells, and a decrease in the expression of genes upregulated by etoposide treatment. This indicates that lamin A has specific roles in the regulation of transcription of genes associated with genome stability. 

Lastly, they analyze the lamin A interactome in the literature and find direct lamin A interactions with some of the factors mentioned above, as well as indirect interactions, for instance, binding of lamin A to factors that bind to promoters of a variety of genes: TP53, MYC, TEAD4, TEAD1, CTCF, YWHAQ, NCOR2, SMAD1, RUNX1, ZBTB7A, MECP2 IGFBP5. 

Overall, this multi-omic study identifies some factors that are directly or indirectly affected by lamin A upon etoposide treatment of cells. While understanding the role of lamins in DNA repair, maintenance of genome integrity, and regulation of gene transcription is important, this study provides relatively limited new knowledge. Only one cell line with highly expressed lamins is utilized and treated with etoposide for a limited amount of time. Whether these specific changes in transcription factors and proteins are a common them in different models is not addressed. Also, some of the -omics correlations are not validated. Overall, the study is a bit superficial, and not clear how the results will allow moving the field forward. I do not believe that the manuscript is appropriate for Cells journal.  

Author Response

REVIEWER 1

Sengupta et al. show that elevated lamins levels in ovarian carcinoma cells compared to immortalized ovarian epithelial cells is associated with alterations in DNA repair machinery, such as elevated DNA repair factors -RAD51, BRCA1, and BRCA2. No elevated basal DNA damage (gH2AX) was observed in tumor or immortalized ovarian cells.

Upon Etoposide (inhibitor of topoisomerase II) treatment, the authors find that high lamins-expressing cells show higher levels of HR mediators such as BRCA1, RAD51, or Ku70. An RNAseq analysis identifies that DEGs include DNA damage/telomere stress-induced senescence, DNA DSB repair, cell cycle regulation, DNA packaging, replication, transcription, recombination, etc. Using network analysis software, they find interactions enriched in DNA damage, telomere stress-induced senescence, meiotic recombination, HDACs and GPCR signaling events.

Finally, used siRNA-mediated KD of lamin A and monitor response of cells to DNA damage (etoposide). Find less BrdU labeled cells in lamin A-depleted cells, and a decrease in the expression of genes upregulated by etoposide treatment. This indicates that lamin A has specific roles in the regulation of transcription of genes associated with genome stability.

Lastly, they analyze the lamin A interactome in the literature and find direct lamin A interactions with some of the factors mentioned above, as well as indirect interactions, for instance, binding of lamin A to factors that bind to promoters of a variety of genes: TP53, MYC, TEAD4, TEAD1, CTCF, YWHAQ, NCOR2, SMAD1, RUNX1, ZBTB7A, MECP2 IGFBP5.

Overall, this multi-omic study identifies some factors that are directly or indirectly affected by lamin A upon etoposide treatment of cells. While understanding the role of lamins in DNA repair, maintenance of genome integrity, and regulation of gene transcription is important, this study provides relatively limited new knowledge. Only one cell line with highly expressed lamins is utilized and treated with etoposide for a limited amount of time. Whether these specific changes in transcription factors and proteins are a common them in different models is not addressed. Also, some of the -omics correlations are not validated. Overall, the study is a bit superficial, and not clear how the results will allow moving the field forward. I do not believe that the manuscript is appropriate for Cells journal. 

Response: We thank the reviewer for the comments. However, we beg to differ with the comment that different models were not addressed. We would like to cite Discher et.al. ,Cell Mol Bioeng.2016 where, two distinct types of ovarian cancer models have been discussed with respect to lamin A expression. The reviewer may please note that we chose the serous carcinoma as opposed to epithelial carcinoma which justifies the choice of OVCAR3. The reason for confining our studies to high grade serous carcinoma ovarian cancer cell line OVCAR3 is the fact that it is aggressively metastatic cancer and the best model to study ovarian cancer drug resistance. The differentially expressed genes do indeed provide newer insights into further investigation of the signaling mechanisms thereby involved in DNA damage repair. Subsequently, proper validation was also performed by qPCR and knockdown studies. Hence, we refute the comment that the results are “superficial, and not clear how the results will allow moving the field forward”.

Reviewer 2 Report

The following points should be carefully considered by the authors and try to improve the review article and make it more rigorous and conclusive.

I would highly recommend that authors should address all the major concerns. As a researcher, I feel that these sections would be very useful for all the readers.

1. Title:

The authors need to revise the title. The abstract should be more precise as it suggests, high expression of Lamins altered the DNA repair machinery specifically in OVCAR3 cells (High grade ovarian serous carcinoma cell line).

2. Abstract section: 

Authors should reframe the abstract and need some minor changes specifically in the last para

3. Introduction Section: 

1. Cite more recent references.

2. Line 52 to 56 “these findings suggest that lamin A/C expression, whether low or high, has a chance of causing metastasis and invasion, given that low levels drive invasion through nuclear deformability, while high levels protect against mechanical forces and thus resist DNA damage-induced cell cycle arrest by assisting the recruitment of DNA damage repair proteins [23]” Authors must provide some more references. Self-citation is not sufficient as this is a very crucial statement.

3.   In an earlier paper Gong et al. and Irianto et al. described that loss of lamin A only associated with metastasis, not lamin C. Authors need to see these articles and provide some explanation. 

a. “Gong, Guanghui, Puxiang Chen, Long Li, Hong Tan, Jun Zhou, Yihong Zhou, Xiaojing Yang, and Xiaoying Wu. "Loss of lamin A but not lamin C expression in epithelial ovarian cancer cells is associated with metastasis and poor prognosis." Pathology-Research and Practice 211, no. 2 (2015): 175-182.”  

b. Irianto, Jerome, Charlotte R. Pfeifer, Irena L. Ivanovska, Joe Swift, and Dennis E. Discher. "Nuclear lamins in cancer." Cellular and molecular bioengineering 9, no. 2 (2016): 258-267.

4.   Line 58-59 “A proteomic investigation was recently conducted in 58 patients with polycystic ovarian disease to identify biomarkers for ovarian cancer.” Authors should give reference here.

5. In Lines 89-92 “It was also reported that DNA breakage decreases the mobility of nucleoplasmic GFP lamin A. Furthermore, lamin A was reported to engage chromatin through phosphorylated H2AX induced by genome damage which was elevated following irradiation. Authors need to work on a literature search and provide some more relevant references. 

6. Since the authors show the association of lamin A with HR and NHEJ-mediated repair mechanisms therefore I am curious to understand if lamin expression alteration affects the DNA damage repair or its mutant expression? Dittmer and Misteli describe “The mutant lamin A protein, named progerin, expressed in HGPS patients has a deletion in the tail domain and consequently remains permanently farnesylated. Cells expressing progerin have delayed recruitment of the repair factor p53-binding protein (53BP1) to sites of DNA damage, show increased levels of the double-stranded break marker γ-H2AX, and are more sensitive to DNA damaging agents”.

a. Dittmer, Travis A., and Tom Misteli. "The lamin protein family." Genome biology 12, no. 5 (2011): 1-14.

b. Liu, Baohua, Jianming Wang, Kui Ming Chan, Wai Mui Tjia, Wen Deng, Xinyuan Guan, Jian-dong Huang, et al. "Genomic instability in laminopathy-based premature aging." Nature medicine 11, no. 7 (2005): 780-785.

7.   Although the authors focus in their manuscript on lamin A expression and their association with HR and NHEJ repair pathways but there is some evidence also discuss that lamin A/C-dependent defect in base excision repair (BER) increased the initiation of multistage carcinogenesis. It would be great if the authors discuss this also and need to explain why the authors focused only on HR and NHEJ. 

a. Maynard, Scott, Guido Keijzers, Mansour Akbari, Michael Ben Ezra, Arnaldur Hall, Marya Morevati, Morten Scheibye-Knudsen, Susana Gonzalo, Jiri Bartek, and Vilhelm A. Bohr. "Lamin A/C promotes DNA base excision repair." Nucleic acids research 47, no. 22 (2019): 11709-11728.

b. Klungland, Arne, Ian Rosewell, Stephan Hollenbach, Elisabeth Larsen, Graham Daly, Bernd Epe, Erling Seeberg, Tomas Lindahl, and Deborah E. Barnes. "Accumulation of premutagenic DNA lesions in mice defective in the removal of oxidative base damage." Proceedings of the National Academy of Sciences 96, no. 23 (1999): 13300-13305. 

8. In line 104. Do authors have to remove the question mark?

4. Material and Methods section: 

1.   Line 130- 137; Treatment with Etoposide and siRNA mediated knockdown of LMNA gene: How authors selected the concentration of siRNA of LMNA? Authors, need to show the concentration-dependent knockdown efficiency of LMNA in supplementary figures.  

2.   Line 138 to 151; I am not sure if this is a typo error. Generally, the secondary antibody working concatenation is in the range of 1:10000 to more. However, the authors used 1:400. Please confirm. Please mention the primary antibody working concentration also.

3.   Line 152-161 Authors need to rewrite in more details about immunofluorescent staining methodology and it will be good if they can provide also the specific dilution of primary and secondary antibodies.

Results Section: 

  1. Line 276-308 “Increased lamin A expression in high-grade ovarian carcinoma cell lines (OVCAR3): 

a.    The authors should revise Figures 1a and b. The representative figure 1a should be a group of three to four cells, not a single-cell image. Also, the image look stretched so my advice is to avoid the extra stretch. What dpi authors have chosen? 

b.    Figure 1b, loading control is not accurate and representative blots have a lot of background noise. Authors need to provide good and clear representative blots and I would like to see the full blot in supplementary figures.

2.    I am wondering if the authors checked the expression of lamin C also.

3.    LINE 302; Magnification: 60X? I think it's 63X. 

4.    Line 309-336; authors observed no change in Ku70 expression. I am wondering if the authors checked the ATM and ATR expression also. It would be more supportive if authors can check their expression as Bradburry et al 2020 also observed the low expression of ATM in OVCAR3 cells. 

5.    Authors need to see this paper “Gee, Mary Ellen, Zahra Faraahi, Aiste McCormick, and Richard J. Edmondson. "DNA damage repair in ovarian cancer: unlocking the heterogeneity." Journal of ovarian research 11, no. 1 (2018): 1-12.” Ellen et al. discussed that Genetic ablation of KU70 and LIG IV restores the survival of PARP-1 deficient cells exposed to DSBs inducing agents. Also, DNA-PK inhibition and depletion have been shown to result in HR function recovery and PARPi resistance in vitro. 

Is it possible that low Ku70 expression induces HR function recovery?

6.    In figure 2 authors should revise the Figure 2a. The representative figure 2a should also be a group of three to four cells, not a single-cell image. Also, I am wondering about BRCA1 expression as it looks like the expression is outside the nucleus means cytoplasmic expression. Is it the case? Also, the authors have shown the BRCA2 expression also, but I struggled to find any discussion about BRCA2 in the text. 

7.    Based on the statistical data analysis, OVCAR3 cells have significantly high expression of BRCA2 compared to BRCA1 (Figure 2a). I am curious to know the reason and it would be great if authors can explain the reason behind this and if any supporting literature supports this finding. 

8.    I am also confused about the lamin A staining in Figure 2a. Images suggest that lamin A expression is low in OVCAR3 cells compared to IOSE cells and this expression profile is totally reversed as authors shown in Figure 1a. 

9.    Figure 2c. Blot quality needs to improve with an accurate loading control. 

10. Did the authors use pure DMSO as a solvent or any specific dilution? 

11. Authors should change the legend in Figure 2 “Mock to sham Control” It makes confusion. 

12. I am not convinced with the image quality and it should be more clear. In Figure 2d. as per my understanding, the representative Image of PCNA suggests the high expression at 48 hr. however, the authors' statistical data is the reverse. The authors have to check their analysis again.

13. Although the blot is clean in Figure 2E sbut till authors have to check the loading control and it's not equal. The major problem is lamin A expression and according to me, there is no difference in lamin A expression between the two groups. Authors have to check this and I am not sure if all the expression profiles in the same blot as the lane, not matches. 

14. Supplementary figure 2a also have a same issue with their loading control.

15. Line 397-492 “Analysis of differentially expressed genes in etoposide-treated OVCAR3 cells:” Authors have nicely shown the differential expression profiles of the genes and the data looks interesting but it should be more clear to readers. It would be great if authors select some specific genes to make some heat maps and correlate with Lamin A expression that how it promotes HR and NHEJ repair machinery. 

16. In Figure 6. I am highly convinced by the author's data regarding the knockdown efficiency by using siLMNA. In Figure 6b authors need to show a representative image like Figure 6a.  

17. I struggle with the data presented in Figures 6b and 6c. The authors have shown that the etoposide treatment in knockdown cells did not change the BrdU expression levels, However, based on the image it gives me the impression that BrdU expression is markedly induced in knockdown cells compared with non-treated cells. The authors need to explain this. 

Discussion Section:

The whole discussion part is very weak. Authors need to rewrite.

Note: 

1.    Reference section needs revisions and authors need to cite more recent papers.

  1. Authors should use GraphPad (Prism) to reanalyze the data. 
  2. All the figure quality need to improve.
  3. Numerous instances in which the spaces between adjacent words are missing. Also, there are several grammatical mistakes. I would suggest that the whole manuscript can be revised thoroughly to improve the manuscript.

Author Response

REVIEWER 2

The following points should be carefully considered by the authors and try to improve the review article and make it more rigorous and conclusive.

I would highly recommend that authors should address all the major concerns. As a researcher, I feel that these sections would be very useful for all the readers.

Response: We thank the reviewer for the constructive comments and suggestions. We have tried to address all the concerns to the best of our knowledge. 

  1. Title:

The authors need to revise the title. The abstract should be more precise as it suggests, high expression of Lamins altered the DNA repair machinery specifically in OVCAR3 cells (High grade ovarian serous carcinoma cell line).

Response: We thank the reviewer for this comment. We have modified the title to “ Elevated levels of lamin A promote HR and NHEJ-mediated repair mechanisms in high-grade ovarian serous carcinoma cell line “ as suggested by the reviewer.

  1. Abstract section:

Authors should reframe the abstract and need some minor changes specifically in the last para

Response: We thank the reviewer for this comment. We have reframed the abstract and modified the last lines (line no 15-19 and 29-31) as suggested by the reviewer.

  1. Introduction Section: 
  2. Cite more recent references.

Response: We thank the reviewer for this comment. Reference nos.  1,2,8,9,25,26,52,53,54,55, 57,63,64 and 83 from the previous version have been replaced with comparatively recent articles. Reference no 10 has been removed. Almost 56 references are of last five years (2017-2022).

2021-2022

2017-2021

2010-2017

2000-2010

1994-2000

1983-1994

10

46

22

27

1

1

  1. Line 52 to 56 “these findings suggest that lamin A/C expression, whether low or high, has a chance of causing metastasis and invasion, given that low levels drive invasion through nuclear deformability, while high levels protect against mechanical forces and thus resist DNA damage-induced cell cycle arrest by assisting the recruitment of DNA damage repair proteins [23]” Authors must provide some more references. Self-citation is not sufficient as this is a very crucial statement.

Response: We thank the reviewer for this comment. We have inserted three more references (Reference no 22,23,24 in the revised manuscript) to support this fact.

  1.  In an earlier paper Gong et al. and Irianto et al. described that loss of lamin A only associated with metastasis, not lamin C. Authors need to see these articles and provide some explanation. 
  2. “Gong, Guanghui, Puxiang Chen, Long Li, Hong Tan, Jun Zhou, Yihong Zhou, Xiaojing Yang, and Xiaoying Wu. "Loss of lamin A but not lamin C expression in epithelial ovarian cancer cells is associated with metastasis and poor prognosis." Pathology-Research and Practice 211, no. 2 (2015): 175-182.”  
  3. Irianto, Jerome, Charlotte R. Pfeifer, Irena L. Ivanovska, Joe Swift, and Dennis E. Discher. "Nuclear lamins in cancer." Cellular and molecular bioengineering 9, no. 2 (2016): 258-267.

Response: We thank the reviewer for this comment. The very specific role of lamin C in caner progression is still very poorly studied and demands greater focus as lamin C has now been reported to dictate genome organisation after mitosis and is associated with several other genetic transactions.  The article referred by the reviewer also concludes that lamin A and not lamin C is a crucial determinant of metastasis in EOC. With these literature in mind, our study emphasizes upon the role of lamin A expression in the neoplastic progressions. As discussed in the referred review by Discher et al, lamin A expression varies with cancer types, origin and stages. Likewise in ovarian cancer scenario, epithelial ovarian cancers are associated with lower levels of lamin A/C while ovarian serous cancers are associated with high levels of the protein. We have restricted our study on high grade ovarian serous cancer (NIH-OVCAR3) where lamin A level is significantly elevated.

  1.  Line 58-59 “A proteomic investigation was recently conducted in 58 patients with polycystic ovarian disease to identify biomarkers for ovarian cancer.”Authors should give reference here.

Response: We thank the reviewer for pointing this out. Suitable reference has been added here (Reference no 25 in the revised manuscript.)

  1. In Lines 89-92 “It was also reported that DNA breakage decreases the mobility of nucleoplasmic GFP lamin A. Furthermore, lamin A was reported to engage chromatin through phosphorylated H2AX induced by genome damage which was elevated following irradiation. Authors need to work on a literature search and provide some more relevant references. 

Response: We thank the reviewer for this comment. We have looked into the literature and three more relevant references have been added (Reference no 38,39 and 40 in the revised manuscript).

  1. Since the authors show the association of lamin A with HR and NHEJ-mediated repair mechanisms therefore I am curious to understand if lamin expression alteration affects the DNA damage repair or its mutant expression? Dittmer and Misteli describe “The mutant lamin A protein, named progerin, expressed in HGPS patients has a deletion in the tail domain and consequently remains permanently farnesylated. Cells expressing progerin have delayed recruitment of the repair factor p53-binding protein (53BP1) to sites of DNA damage, show increased levels of the double-stranded break marker γ-H2AX, and are more sensitive to DNA damaging agents”.
  2. Dittmer, Travis A., and Tom Misteli. "The lamin protein family." Genome biology 12, no. 5 (2011): 1-14.
  3. Liu, Baohua, Jianming Wang, Kui Ming Chan, Wai Mui Tjia, Wen Deng, Xinyuan Guan, Jian-dong Huang, et al. "Genomic instability in laminopathy-based premature aging." Nature medicine 11, no. 7 (2005): 780-785.

Response: We thank the reviewer for this comment. There are various studies on perturbation of DNA damage repair machineries by introducing mutations in lamin A in laminopathy models

[ Gonzalez-Suarez I, Gonzalo S (2010) Nurturing the genome: A-type lamins preserve genomic stability. Nucleus 1: 129-135. DOI 10.4161/nucl.1.2.10797 ]

[ Manju K, Muralikrishna B, Parnaik VK (2006) Expression of disease-causing lamin A mutants impairs the formation of DNA repair foci. J Cell Sci 119: 2704-2714. DOI 10.1242/jcs.03009 ]

[ Huang S, Risques RA, Martin GM, Rabinovitch PS, Oshima J (2008) Accelerated telomere shortening and replicative senescence in human fibroblasts overexpressing mutant and wild-type lamin A. Exp Cell Res 314: 82-91. DOI 10.1016/j.yexcr.2007.08.004). ]

Similarly, studies have also found improper recruitment of DNA Damage repair proteins like RAD50, XPA at the DSBs, reduced transcription of DNA damage repair proteins /9RAD51, BRCA1) , altered localisation (pRb, ING1, 53BP1) and decreased stability of the repair proteins (53BP1) leading to faulty damage repair mechanism due to aberrant expressions of lamin A.

[ Redwood AB, Perkins SM, Vanderwaal RP, Feng Z, Biehl KJ, Gonzalez-Suarez I, Morgado-Palacin L, Shi W, Sage J, Roti-Roti JL, et al. (2011) A dual role for A-type lamins in DNA double-strand break repair. Cell Cycle 10: 2549-2560. DOI 10.4161/cc.10.15.16531 ]

[ Nitta RT, Jameson SA, Kudlow BA, Conlan LA, Kennedy BK (2006) Stabilization of the retinoblastoma protein by A-type nuclear lamins is required for INK4A-mediated cell cycle arrest. Mol Cell Biol 26: 5360-5372. DOI 10.1128/MCB.02464-05 ]

[ Han X, Feng X, Rattner JB, Smith H, Bose P, Suzuki K, Soliman MA, Scott MS, Burke BE, Riabowol K (2008) Tethering by lamin A stabilizes and targets the ING1 tumour suppressor. Nat Cell Biol 10: 1333-1340. DOI 10.1038/ncb1792 ]

[ Gonzalez-Suarez I, Redwood AB, Perkins SM, Vermolen B, Lichtensztejin D, Grotsky DA, Morgado-Palacin L, Gapud EJ, Sleckman BP, Sullivan T, et al. (2009) Novel roles for A-type lamins in telomere biology and the DNA damage response pathway. EMBO J 28: 2414-2427. DOI 10.1038/emboj.2009.196 ]

[Gonzalez-Suarez I, Redwood AB, Grotsky DA, Neumann MA, Cheng EH, Stewart CL, Dusso A, Gonzalo S (2011) A new pathway that regulates 53BP1 stability implicates cathepsin L and vitamin D in DNA repair. EMBO J 30: 3383-3396. DOI 10.1038/emboj.2011.225. ]

[Sengupta D, Mukhopadhyay A, Sengupta K. Emerging roles of lamins and DNA damage repair mechanisms in ovarian cancer. Biochem Soc Trans. 2020 Oct 30;48(5):2317-2333. doi: 10.1042/BST20200713. PMID: 33084906.]

In summary, absence or deficiency of functional lamin A (either by mutation or by aberrant expression) renders a cell more prone to genome damage.

  1.  Although the authors focus in their manuscript on lamin A expression and their association with HR and NHEJ repair pathways but there is some evidence also discuss that lamin A/C-dependent defect in base excision repair (BER) increased the initiation of multistage carcinogenesis. It would be great if the authors discuss this also and need to explain why the authors focused only on HR and NHEJ. 
  2. Maynard, Scott, Guido Keijzers, Mansour Akbari, Michael Ben Ezra, Arnaldur Hall, Marya Morevati, Morten Scheibye-Knudsen, Susana Gonzalo, Jiri Bartek, and Vilhelm A. Bohr. "Lamin A/C promotes DNA base excision repair." Nucleic acids research 47, no. 22 (2019): 11709-11728.
  3. Klungland, Arne, Ian Rosewell, Stephan Hollenbach, Elisabeth Larsen, Graham Daly, Bernd Epe, Erling Seeberg, Tomas Lindahl, and Deborah E. Barnes. "Accumulation of premutagenic DNA lesions in mice defective in the removal of oxidative base damage." Proceedings of the National Academy of Sciences 96, no. 23 (1999): 13300-13305. 

Response: We thank the reviewer for this comment. Roles of lamin A in all types of DNA damage repair machineries like HR, NHEJ, NER, BER and Mismatch Repair due to altered expression of lamin A have been discussed here.

  1. Homologous recombination (Redwood AB, Perkins SM, Vanderwaal RP, Feng Z, Biehl KJ, Gonzalez-Suarez I, Morgado-Palacin L, Shi W, Sage J, Roti-Roti JL, Stewart CL, Zhang J, Gonzalo S. A dual role for A-type lamins in DNA double-strand break repair. Cell Cycle. 2011 Aug 1;10(15):2549-60. doi: 10.4161/cc.10.15.16531. Epub 2011 Aug 1. PMID: 21701264; PMCID: PMC3180193.),
  2. Non homologous End Joining (Gibbs-Seymour I, Markiewicz E, Bekker-Jensen S, Mailand N, Hutchison CJ. Lamin A/C-dependent interaction with 53BP1 promotes cellular responses to DNA damage. Aging Cell. 2015;14(2):162-169. doi:10.1111/acel.12258),
  3. Base excision repair (Maynard S, Keijzers G, Akbari M, et al. Lamin A/C promotes DNA base excision repair. Nucleic Acids Res. 2019;47(22):11709-11728. doi:10.1093/nar/gkz912),
  4. Mismatch repair (Tiwari V, Wilson DM 3rd. DNA Damage and Associated DNA Repair Defects in Disease and Premature Aging. Am J Hum Genet. 2019 Aug 1;105(2):237-257. doi: 10.1016/j.ajhg.2019.06.005. PMID: 31374202; PMCID: PMC6693886.,) and
  5. Nucleotide excision repair (Liu Y, Wang Y, Rusinol AE, et al. Involvement of xeroderma pigmentosum group A (XPA) in progeria arising from defective maturation of prelamin A. FASEB J. 2008;22(2):603-611. doi: 10.1096/fj.07-8598com)

To specifically address this alteration in expression in context of progression of high grade ovarian serous cancer, we selectively focussed on two common mechanisms namely Homologous recombination and Non homologous end joining, as defective HR and NHEJ have been found in almost 50% of ovarian cancer cases where as association with the other repair processes are 0.8-8% (Gee ME, Faraahi Z, McCormick A, Edmondson RJ. DNA damage repair in ovarian cancer: unlocking the heterogeneity. J Ovarian Res. 2018 Jun 20;11(1):50. doi: 10.1186/s13048-018-0424-x. PMID: 29925418; PMCID: PMC6011341.).

  1. In line 104. Do authors have to remove the question mark?

Response: We thank the reviewer for pointing out the mistake. We have removed the question mark.

  1. Material and Methods section: 
  2.  Line 130- 137; Treatment with Etoposide and siRNA mediated knockdown of LMNA gene:How authors selected the concentration of siRNA of LMNA? Authors, need to show the concentration-dependent knockdown efficiency of LMNA in supplementary figures.  

Response: We thank the reviewer for this comment. The sequences of siRNA, the concentration and the protocol for transfection were finalised following the method mentioned in the article by Harada, T., et al., JCB, 2014  (Harada, T., et al., Nuclear lamin stiffness is a barrier to 3D migration, but softness can limit survival. J Cell Biol, 2014. 204(5): p. 669-82.). Initially we checked the exact concentration of siRNA complex mentioned in the article which is 30nM and another concentration which is slightly higher (50 nM) (the blot has been inserted as supplementary Figure 5 in supporting information 1). Since the extent of knock down was not satisfactory with 30 nM or 50 nM., we further checked with one more concentration (60nM) which exhibited satisfactory knockdown efficiency. Hence, 60nM was used for all the knock down experiments and the blot is shown in Figure 6A.

  1.  Line 138 to 151;I am not sure if this is a typo error. Generally, the secondary antibody working concatenation is in the range of 1:10000 to more. However, the authors used 1:400. Please confirm. Please mention the primary antibody working concentration also.

Response: We thank the reviewer for this comment. The specific dilutions for all the primary antibodies used in this article for immunostaining and western blot are mentioned in Supplementary Table 2 in Supporting Information file 1. The dilution for the secondary antibody is 1:400 which falls in the recommended dilution range for optimised activity for this specific secondary antibody “Goat anti-Mouse IgG (H+L) Secondary Antibody, HRP” : Cat no 32430. (https://www.thermofisher.com/order/genome-database/dataSheetPdf?producttype=antibody&productsubtype=antibody_secondary&productId=32430&version=260)

  1.  Line 152-161 Authors need to rewrite in more details about immunofluorescent staining methodology and it will be good if they can provide also the specific dilution of primary and secondary antibodies.

 Response: We thank the reviewer for this comment. We have revised and elaborately discussed the immunofluorescent staining methodology. Specific dilutions for all the primary antibodies used in this article for immunostaining and western blot are mentioned in Supplementary Table 2 in Supporting Information file 1. The dilution for the secondary antibody is 1:400.

Results Section: 

  1. Line 276-308 “Increased lamin A expression in high-grade ovarian carcinoma cell lines (OVCAR3): 
  1. The authors should revise Figures 1a and b. The representative figure 1a should be a group of three to four cells, not a single-cell image. Also, the image look stretched so my advice is to avoid the extra stretch. What dpi authors have chosen? 
  2. Figure 1b, loading control is not accurate and representative blots have a lot of background noise. Authors need to provide good and clear representative blots and I would like to see the full blot in supplementary figures.

Response: We thank the reviewer for this comment. We have revised the figure with a greater number of lamin A and lamin B-stained nuclei in one field for both the cell lines. We also have eliminated extra stretching. The image is of 300 dpi. Full blots have been added in the Supporting Information file 1

  1. I am wondering if the authors checked the expression of lamin C also.

Response: We thank the reviewer for this comment. The antibody used in this study for lamin A is Anti-Lamin A (C-terminal) antibody produced in rabbit (L1293)(Sigma -Aldrich) which shows only one band for lamin A in 74kD. So, we have not checked the expression of lamin C

  1. LINE 302; Magnification: 60X? I think it's 63X. 

Response: We thank the reviewer for this comment. We are extremely sorry and apologise for the mistake. We have inserted a correction in the Immunocytochemistry and Data analysis section of material and methods (line no 158) that imaging was also performed in 60X water immersion objectives in NIKON TiE inverted research microscope. These images are captured in 60X.

  1. Line 309-336; authors observed no change in Ku70 expression. I am wondering if the authors checked the ATM and ATR expression also. It would be more supportive if authors can check their expression as Bradburry et al 2020 also observed the low expression of ATM in OVCAR3 cells. 

Response: We thank the reviewer for this comment. The level of Ku70 remains the same in IOSE and OVCAR3 which was also established earlier by Langland et al, 2010. We have added the reference in line no 299. As, Ku70 level does not change between OVCAR3 and IOSE, We have not checked the expressions of ATM and ATR as reported earlier

[Tomimatsu N, Tahimic CG, Otsuki A, Burma S, Fukuhara A, Sato K, Shiota G, Oshimura M, Chen DJ, Kurimasa A. Ku70/80 modulates ATM and ATR signaling pathways in response to DNA double strand breaks. J Biol Chem. 2007 Apr 6;282(14):10138-45. doi: 10.1074/jbc.M611880200. Epub 2007 Feb 1. PMID: 17272272.]

  1. Authors need to see this paper “Gee, Mary Ellen, Zahra Faraahi, Aiste McCormick, and Richard J. Edmondson. "DNA damage repair in ovarian cancer: unlocking the heterogeneity." Journal of ovarian research 11, no. 1 (2018): 1-12.” Ellen et al. discussed that Genetic ablation of KU70 and LIG IV restores the survival of PARP-1 deficient cells exposed to DSBs inducing agents. Also, DNA-PK inhibition and depletion have been shown to result in HR function recovery and PARPi resistance in vitro. 

Is it possible that low Ku70 expression induces HR function recovery?

Response: We thank the reviewer for this comment. We have not observed any change in Ku70 expression. But the level of Ku70 in OVCAR3 has not been found to be low which has been reported earlier as well by Langland et al, 2010. We have discussed it in details in the discussion section between lines 617-627.

“Researchers have performed CGH profiling on the panel of ovarian cell lines to see if gross amplifications or deletions in these locations associated with relative X-ray sensitivity exist, because Ku70 and DNA-PK exist on chromosomes that are frequently found to be abnormal in ovarian malignancies [67]. However, no significant correlations were observed. Interestingly, in accordance with our finding, this study also could find no difference in Ku70 protein expression between OVCAR3 and IOSE cells [67]. Although, there were changes in the copy number of DNA-PKs, Ku70 and Ku80 which did not correlate with radiosensitivity [67].

  1. In figure 2 authors should revise the Figure 2a. The representative figure 2a should also be a group of three to four cells, not a single-cell image. Also, I am wondering about BRCA1 expression as it looks like the expression is outside the nucleus means cytoplasmic expression. Is it the case? Also, the authors have shown the BRCA2 expression also, but I struggled to find any discussion about BRCA2 in the text. “

Response: We thank the reviewer for this comment. We have revised the figure and added merged images showing the localisation of γH2AX, Ku 70, BRCA1 and BRCA2 along with DAPI to mark the nucleus. However, we would like to clarify that the panel with these set of confocal images was to revalidate the altered expression of the repair proteins in the cell lines which had been established by blot and qPCR results. We could not find any newer insights regarding any change in localisation of the proteins between the two cell lines. BRCA1 expression has been noticed in the cytosol as well as in the nucleus, however,  cytosolic localisation of BRCA1 in OVCAR3 has been previously reported by several groups .

Wang H, Shao N, Ding QM, Cui J, Shyam E, Reddy P & Rao VN (1997) BRCA1 proteins are transported to the nucleus in the absence of serum and splice variants BRCA1a, BRCA1b are tyrosine phsophoproteins that associate with E2F, cyclins and cyclin dependent kinases. Oncogene 15, 143– 157.

Qin Y, Xu J, Aysola K, Oprea G, Reddy A, Matthews R, Okoli J, Cantor A, Grizzle WE, Partridge EE, Reddy ES, Landen C, Rao VN. BRCA1 proteins regulate growth of ovarian cancer cells by tethering Ubc9. Am J Cancer Res. 2012;2(5):540-8. Epub 2012 Aug 20. PMID: 22957306; PMCID: PMC3433105.

We have revised and discussed regarding BRCA2 expression in the discussion part from line no 605-6017.

“ BRCA1/2 are among the genes that are essential for homologous recombination repair. BRCA2 operates almost solely in homologous recombination, in contrast to BRCA1, which has multiple activities[103, 104]. Studies have found that ovarian cancer tissues have higher transcriptome-level BRCA1/2 expression which is in line with our finding[105]. The faster rate of proliferation in malignant tissues, along with genetic instability needs a greater requirement for DNA damage repair which may be the root cause of these changes[105]. Accordingly, high-grade cancers also showed greater BRCA1/2 expression in earlier studies[105]. Gudas et al. provide credence to this idea by proposing that the proliferation of breast cancer cells is indirectly responsible for the elevation of BRCA1 expression by steroid hormones[105]. It is to be noted that OVCAR3 cells did not have any mutations in known HR repair genes, but deletions were reported in several HRR-related genes including BRCA2.  However, in BRCA2 wild-type cases of ovarian cancer, a high level of BRCA2 mRNA expression is one of the determinants of chemoresistance [68, 105]. “

  1. Based on the statistical data analysis, OVCAR3 cells have significantly high expression of BRCA2 compared to BRCA1 (Figure 2a). I am curious to know the reason and it would be great if authors can explain the reason behind this and if any supporting literature supports this finding. 

Response: We thank the reviewer for this comment. We have checked the literature and could find this article referred here.

Tsibulak I, Wieser V, Degasper C, Shivalingaiah G, Wenzel S, Sprung S, Lax SF, Marth C, Fiegl H, Zeimet AG. BRCA1 and BRCA2 mRNA-expression prove to be of clinical impact in ovarian cancer. Br J Cancer. 2018 Sep;119(6):683-692. doi: 10.1038/s41416-018-0217-4. Epub 2018 Aug 15. PMID: 30111871; PMCID: PMC6173779.

This paper has shown higher BRCA2 mRNA expression than BRCA1 in high grade ovarian cancer patients. We could not find any genomic studies of BRCA1 and BRCA2 in OVCAR3 cell line model. We have discussed the key finding of this paper relevant to our study in discussion section between lines 605-617.

“BRCA1/2 are among the genes that are essential for homologous recombination repair. BRCA2 operates almost solely in homologous recombination, in contrast to BRCA1, which has multiple activities[103, 104]. Studies have found that ovarian cancer tissues have higher transcriptome-level BRCA1/2 expression which is in line with our finding[105]. The faster rate of proliferation in malignant tissues, along with genetic instability needs a greater requirement for DNA damage repair which may be the root cause of these changes[105]. Accordingly, high-grade cancers also showed greater BRCA1/2 expression in earlier studies[105]. Gudas et al. provide credence to this idea by proposing that the proliferation of breast cancer cells is indirectly responsible for the elevation of BRCA1 expression by steroid hormones[105]. It is to be noted that OVCAR3 cells did not have any mutations in known HR repair genes, but deletions were reported in several HRR-related genes including BRCA2.  However, in BRCA2 wild-type cases of ovarian cancer, a high level of BRCA2 mRNA expression is one of the determinants of chemoresistance [68, 105].”

  1. I am also confused about the lamin A staining in Figure 2a. Images suggest that lamin A expression is low in OVCAR3 cells compared to IOSE cells and this expression profile is totally reversed as authors shown in Figure 1a. 

Response: We thank the reviewer for this comment. We clarify that we had distinctly shown an elevated expression of lamin A in OVCAR3 compared to IOSE. We quantified the same using ImageJ mentioned in the method section in details. We would like to draw the attention of the reviewer to figure 2a where the top panel exhibiting lower expression lamin A corresponds to IOSE and the bottom panel showing elevated expression of lamin A corresponds to OVCAR3. Hence, there is nothing to revise in this figure.

  1. Figure 2c. Blot quality needs to improve with an accurate loading control. 

Response: We thank the reviewer for this comment. We have repeated the experiments and revised the blot for accurate loading control as suggested by the reviewer.

  1. Did the authors use pure DMSO as a solvent or any specific dilution? 

Response: We thank the reviewer for this comment. DMSO was used as solvent for etoposide and was introduced in the control experiments in the similar volume as that of the required volume of etoposide.

  1. Authors should change the legend in Figure 2 “Mock to sham Control” It makes confusion. 

Response: We thank the reviewer for this comment. The reviewer might have mistaken the figure 3 as figure 2. In figure 3, we have changed Mock to sham control as and where required.

  1. I am not convinced with the image quality and it should be more clear. In Figure 2d. as per my understanding, the representative Image of PCNA suggests the high expression at 48 hr. however, the authors' statistical data is the reverse. The authors have to check their analysis again.

Response: We thank the reviewer for this comment. The bar diagram in figure 3D (PCNA staining) denotes the relative change in PCNA level in 24 and 48 hours of etoposide treatment with respect to DMSO treated cells. We could find that PCNA fluorescence was higher after 24 hours of etoposide treatment which reduced slightly in the 48 hours timepoint. This reduction has been analysed and found to be statistically significant. This experiment was performed initially to optimise the duration of etoposide treatment. All the experiments afterward have been performed considering 24 hours  of etoposide treatment.

  1. Although the blot is clean in Figure 2E sbut till authors have to check the loading control and it's not equal. The major problem is lamin A expression and according to me, there is no difference in lamin A expression between the two groups. Authors have to check this and I am not sure if all the expression profiles in the same blot as the lane, not matches. 

Response: We thank the reviewer for this comment. Lamin A level was not found to be altered after treating OVCAR3 cells with etoposide, which has been validated by blot and immunofluorescence. Also, the RNA seq data of etoposide treated OVCAR3 reveals that LMNA did not appear in the list of 205 differentially expressed genes. So, lamin A level was unaltered following etoposide treatment

  1. Supplementary figure 2a also have a same issue with their loading control.

Response: We thank the reviewer for this comment. We have repeated the experiment and revised the figure.

  1. Line 397-492 “Analysis of differentially expressed genes in etoposide-treated OVCAR3 cells:” Authors have nicely shown the differential expression profiles of the genes and the data looks interesting but it should be more clear to readers. It would be great if authors select some specific genes to make some heat maps and correlate with Lamin A expression that how it promotes HR and NHEJ repair machinery. 

Response: We thank the reviewer for this comment. We have revised Figure 6 and inserted a heatmap as Figure 6F.

  1. In Figure 6. I am highly convinced by the author's data regarding the knockdown efficiency by using siLMNA. In Figure 6b authors need to show a representative image like Figure 6a.  

Response: We thank the reviewer for this comment. Lamin A level was not found to be altered after treating OVCAR3 cells with etoposide, which has been validated by blot and immunofluorescence. Also, the RNA seq data of etoposide treated OVCAR3, reveals that LMNA did not appear in the list of 205 differentially expressed genes. So, lamin A level was unaltered following etoposide treatment and is only reduced after siRNA knockdown. So, another representative image of knock down in etoposide treated OVCAR3 cells would provide no added information.

  1. I struggle with the data presented in Figures 6b and 6c. The authors have shown that the etoposide treatment in knockdown cells did not change the BrdU expression levels, However, based on the image it gives me the impression that BrdU expression is markedly induced in knockdown cells compared with non-treated cells. The authors need to explain this. 

Response: We thank the reviewer for this comment. The image in Figure 6B shows a prominent drop in the number of BrdU positive cells in LMNA knock down OVCAR3 cells and the difference in the number of BrdU-positive cells was insignificant between the etoposide-treated and untreated counterparts of LMNA-deficient OVCAR3 which indicates that first of all etoposide triggers S phase accumulation (Nam, C., K. Doi, and H. Nakayama, Etoposide induces G2/M arrest and apoptosis in neural progenitor cells via DNA damage and an ATM/p53-related pathway. Histol Histopathol, 2010. 25(4): p. 485-93) and secondly lamin A drives the process positively. This observation has been revalidated by PCNA staining in etoposide treated OVCAR3 cells before and after knockdown. PCNA fluorescence was much lower in etoposide treated LMNA knock down OVCAR3 cells (Supplementary Figure 6)

Discussion Section:

The whole discussion part is very weak. Authors need to rewrite.

 Response: We thank the reviewer for this comment. We have suitably modified the discussion.

Note: 

  1. Reference section needs revisions and authors need to cite more recent papers.

  1. Authors should use GraphPad (Prism) to reanalyze the data

  1. All the figure quality need to improve.

  1. Numerous instances in which the spaces between adjacent words are missing. Also, there are several grammatical mistakes. I would suggest that the whole manuscript can be revised thoroughly to improve the manuscript.

 Response: We thank the reviewer for this comment. We have addressed all the above-mentioned points.

Reviewer 3 Report

Overall comments:

In several cancer cell types, there have been reports of altered lamin expression and although nuclear lamin is shown to affect multiple cellular processes such as cell invasion, recruitment of DNA damage repair proteins, the mechanism of how its expression in cells could contribute to cancer progression is not clearly understood. Using an ovarian cancer model system and etoposide as a DNA damaging agent, this study aims to systematically probe the molecular interactors of lamin A and derive conclusions on how its expression can affect ovarian serous cancer biology. The experimental setup is well validated using two ovarian cancer cell lines with significantly different levels of lamin A expression. The authors use RNA seq to study global gene expression alterations upon DNA damage and in the context of high lamin A expression. With the help of several protein-protein interactome analysis platforms, they successfully highlight the possible molecular interactors of lamin A, and the various pathways involved to map out a functional circuitry. While these possible protein interactions deserve orthologous biochemical validation in future studies, the authors provide a plausible explanation for how these putative interactions might impact various pathways affected by lamin knockdown in OVCAR3 to conceptualize how lamin A could contribute to maintaining genomic stability, cell proliferation, exhibit chemoresistance. Introduction is well-written with adequate background information about how the Lamins are implicated in cancer progression. The manuscript overall contributes useful information on lamin A interactome, that will be useful to the field; however, it would benefit from addressing several experimental concerns as listed below.

Major points:

·      While the paper identifies putative lamin interactors, it lacks validation for any of them. The authors mention 5 of the 12 representative interactors to be known interactors and cite adequate references. However, for the new interacting partners, a minimum requirement for any scientific claim would be a co-IP with lamin A, exhibiting specificity with a proper control.

·      Immunoblots in Figure 3E, 6A show only one band corresponding to lamin A (whereas that of 1B looks like there might be two bands if the gel was run longer). Lamin antibody should normally detect a slightly lower band of Lamin C as well (also refer to minor point below regarding mention of catalog numbers for antibodies). If there is any discrepancy, the antibody should be validated using a KO or KD of lamin A.

·      The introduction includes recent reports on how lamin expression in altered in cancerous cells, however, to justify the focus on lamin A for this study, may be mention - while lamin A/C and lamin B are altered in ovarian cancer (Bengtsson 2007, Capo-chichi 2011), only lamin A/C is known to alter in ovarian serous cancer (Wang 2009).

·      The authors validated lamin protein and mRNA levels in the two cell lines and found OVCAR3 to have higher levels of both A and B type lamins (when compared to IOSE). However, they state that differences in lamin B are statistically insignificant (Figure 1C). This sounds misleading, for if we look at that figure, there seems to be an even bigger difference in lamin B mRNA levels than that of lamin A. A more elaborate understanding of the statistic done here is needed. This figure also lacks a y-axis label, the addition of which would help address this discrepancy.

·      The authors find HR proteins BRCA1 and Rad51 to be elevated in OVCAR3 but they mention in text (lines 325-326) that expression of Ku was similar and then go on to explain how there’s a discrepancy in Ku expression between studies and how this may not be the appropriate determinant of DNA damage repair. However, in figure 2B and 2D, they show statistically significant lower levels of Ku in OVCAR3. This needs to be explained in text.

·      In Figure 1A, is there an increase in fluorescence at the nuclear rim specifically? Or does OVCAR3 have higher anti-lamin A nucleoplasmic signal? In fact, the line plot for lamin B fluorescence intensity indicates higher levels in OVCAR3 (grey) than the IOSE (blue) at the edges of the plot, which one would assume, denotes the nuclear rim. However, if we look at the same plot for lamin A, OVCAR3 has a higher intensity closer to nucleoplasm than at the nuclear rim. An explanation for this in the results section is crucial to our understanding of this study. A quantification specifically at the nuclear rim would provide a more concrete answer.

·      Several graphs cannot be interpreted- Figure 1C and 2D should have y-axis labels. In Figure 1C, 2D, 6E, some of the statistics indicate that the differences are not significant but they look way higher and with tighter error bars than an adjoining data point (where it mentions as significant), and thus need further explanation and/or a revisit at the statistical analysis done. There needs to be a proper mention of number of biological/technical replicates, number of cells used for each data point along with a clear description of the statistics used.

Minor points:

·      The bottom immunoblot in Figure 1B could benefit from additional repetitions of the blot for comparison, as it is clearly unequally loaded

·      Assuming in all their etoposide experiments, the treatment was done for 24h (as mentioned in the Methods section), why was a 48h time point used when examining the differences in PCNA levels (Figure 3D)? The rationale deserves an explanation in the text.

·      The list of differentially expressed genes and corresponding pathways (422-456) would be easier to assimilate in a table format.

·      In Suppl information file 1, the list of antibodies should also include catalog numbers along with the company it was purchased from.

·      While detecting more S-phase nuclei in Lamin A downregulated OVCAR3, did the authors look at if PCNA levels were altered? They already had done this for etoposide treated OVCAR3, Figure 3D and it would support their claim if PCNA levels altered in lamin1 knockdown.

Author Response

REVIEWER 3

Overall comments:

In several cancer cell types, there have been reports of altered lamin expression and although nuclear lamin is shown to affect multiple cellular processes such as cell invasion, recruitment of DNA damage repair proteins, the mechanism of how its expression in cells could contribute to cancer progression is not clearly understood. Using an ovarian cancer model system and etoposide as a DNA damaging agent, this study aims to systematically probe the molecular interactors of lamin A and derive conclusions on how its expression can affect ovarian serous cancer biology. The experimental setup is well validated using two ovarian cancer cell lines with significantly different levels of lamin A expression. The authors use RNA seq to study global gene expression alterations upon DNA damage and in the context of high lamin A expression. With the help of several protein-protein interactome analysis platforms, they successfully highlight the possible molecular interactors of lamin A, and the various pathways involved to map out a functional circuitry. While these possible protein interactions deserve orthologous biochemical validation in future studies, the authors provide a plausible explanation for how these putative interactions might impact various pathways affected by lamin knockdown in OVCAR3 to conceptualize how lamin A could contribute to maintaining genomic stability, cell proliferation, exhibit chemoresistance. Introduction is well-written with adequate background information about how the Lamins are implicated in cancer progression. The manuscript overall contributes useful information on lamin A interactome, that will be useful to the field; however, it would benefit from addressing several experimental concerns as listed below.

Response: We thank the reviewer for the constructive comments and suggestions. We have tried to address all the concerns to the best of our knowledge. 

Major points:

  • While the paper identifies putative lamin interactors, it lacks validation for any of them. The authors mention 5 of the 12 representative interactors to be known interactors and cite adequate references. However, for the new interacting partners, a minimum requirement for any scientific claim would be a co-IP with lamin A, exhibiting specificity with a proper control.

Response: We thank the reviewer for this comment. We beg to state that at this point it would not be possible for us to validate the claims by co-IP due to paucity of funds and time. However, we believe that we have validated the same by qPCR. We would be obliged if that can be taken into account to establish the veracity of the claim.

  • Immunoblots in Figure 3E, 6A show only one band corresponding to lamin A (whereas that of 1B looks like there might be two bands if the gel was run longer). Lamin antibody should normally detect a slightly lower band of Lamin C as well (also refer to minor point below regarding mention of catalog numbers for antibodies). If there is any discrepancy, the antibody should be validated using a KO or KD of lamin A.

Response: We thank the reviewer for this comment. The antibody used for lamin A is Anti-Lamin A (C-terminal) antibody produced in rabbit (L1293) (Sigma Aldrich) which is supposed to show one band at 74kD as per the datasheet and product details. 

(https://www.sigmaaldrich.com/IN/en/product/sigma/l1293)

We also have similar observation and have provided the figure at the suitable place (Figure 1B)

  • The introduction includes recent reports on how lamin expression in altered in cancerous cells, however, to justify the focus on lamin A for this study, may be mention - while lamin A/C and lamin B are altered in ovarian cancer (Bengtsson 2007, Capo-chichi 2011), only lamin A/C is known to alter in ovarian serous cancer (Wang 2009).

Response: We thank the reviewer for this comment. We had already mentioned the studies by Capo-chichi et al 2011 and Wand et al 2009 in reference number 16 and 18. We have mentioned the study by Bengtsson et al 2007 in the revised manuscript (Reference no 79).

  • The authors validated lamin protein and mRNA levels in the two cell lines and found OVCAR3 to have higher levels of both A and B type lamins (when compared to IOSE). However, they state that differences in lamin B are statistically insignificant (Figure 1C). This sounds misleading, for if we look at that figure, there seems to be an even bigger difference in lamin B mRNA levels than that of lamin A . A more elaborate understanding of the statistic done here is needed. This figure also lacks a y-axis label, the addition of which would help address this discrepancy.

Response: We thank the reviewer for this comment. The discrepancy for the results regarding lamin B mainly involves lack of reproducibility. From the blot and immunofluorescence data, difference between IOSE lamin B level and OVCAR3 lamin B levels were less than that of the difference in lamin A levels between the two cell lines. But in RT-PCR experiments, lamin B fold changes were highly varying in experimental replicates. So, the data was not reliable to proceed with.

As suggested by the reviewer, we have added y axis labels in the all the bar charts provided in the manuscript. We apologise for the inadvertent manner in the former manuscript.

  • The authors find HR proteins BRCA1 and Rad51 to be elevated in OVCAR3 but they mention in text (lines 325-326) that expression of Ku was similar and then go on to explain how there’s a discrepancy in Ku expression between studies and how this may not be the appropriate determinant of DNA damage repair. However, in figure 2B and 2D, they show statistically significant lower levels of Ku in OVCAR3. This needs to be explained in text.

Response: We thank the reviewer for this comment. The level of Ku70 remains the same in IOSE and OVCAR3 which was also established earlier by Langland et al, 2010. We have added the reference in line no 299. We have also discussed regarding changes in ku70 protein expression and change in their gene copy number in the discussion part from line number 617-627.

“Researchers have performed CGH profiling on the panel of ovarian cell lines to see if gross amplifications or deletions in these locations associated with relative X-ray sensitivity exist, because Ku70 and DNA-PK exist on chromosomes that are frequently found to be abnormal in ovarian malignancies [67]. However, no significant correlations were observed. Interestingly, in accordance with our finding, this study also could find no difference in Ku70 protein expression between OVCAR3 and IOSE cells [67]. Although, there were changes in the copy number of DNA-PKs, Ku70 and Ku80 which did not correlate with radiosensitivity [67].”

  • In Figure 1A, is there an increase in fluorescence at the nuclear rim specifically? Or does OVCAR3 have higher anti-lamin A nucleoplasmic signal? In fact, the line plot for lamin B fluorescence intensity indicates higher levels in OVCAR3 (grey) than the IOSE (blue) at the edges of the plot, which one would assume, denotes the nuclear rim. However, if we look at the same plot for lamin A, OVCAR3 has a higher intensity closer to nucleoplasm than at the nuclear rim. An explanation for this in the results section is crucial to our understanding of this study. A quantification specifically at the nuclear rim would provide a more concrete answer.

Response: We thank the reviewer for this comment. The explanation has been added in the result section from line no 269-273.

“Elevation in lamin A and lamin B levels are visibly clear both in the nuclear rim and nucleoplasm in OVCAR3 nuclei with respect to IOSE, although abundance of nucleoplasmic lamin A is more prominent in OVCAR3 nuclei.This is even more intriguing as interaction with the transcription factors are mostly carried out by the nucleoplasmic lamin A”

 Furthermore the scheme of quantification of lamin A and lamin B in the nuclear rim has been added in supplementary figure 1 in the revised manuscript.

  • Several graphs cannot be interpreted- Figure 1C and 2D should have y-axis labels. In Figure 1C, 2D, 6E, some of the statistics indicate that the differences are not significant but they look way higher and with tighter error bars than an adjoining data point (where it mentions as significant), and thus need further explanation and/or a revisit at the statistical analysis done. There needs to be a proper mention of number of biological/technical replicates, number of cells used for each data point along with a clear description of the statistics used.

Response: We thank the reviewer for this comment. We apologise for the inadvertent error in marking the y axis and have added y axis labels in the all the bar charts provided in the manuscript, as suggested by the reviewer. Immunofluorescence, western blot and qPCR experiments were performed at least 6 times (mentioned in line no 142, 164, 206 in the revised manuscript). Around 250 nuclei were used for quantification of fluorescence intensities. Comet assay experiments were performed in triplicates. 10-15 cells were analyzed in each field. 20 such fields were studied for each sample (mentioned in line no. 180-181). Experimental replicates for BrdU and MTT assay were already mentioned in line no 190-191, 198. Statistical significance was determined by the 2-tailed Student’s t-test with a value of p < 0.05 and was already mentioned in line no 246.

Minor points:

  • The bottom immunoblot in Figure 1B could benefit from additional repetitions of the blot for comparison, as it is clearly unequally loaded

Response: We thank the reviewer for pointing this out. We have repeated the blots and hence revised the figures as suggested by the reviewer.

  • Assuming in all their etoposide experiments, the treatment was done for 24h (as mentioned in the Methods section), why was a 48h time point used when examining the differences in PCNA levels (Figure 3D)? The rationale deserves an explanation in the text.

Response: We thank the reviewer for this comment. The bar diagram in figure 3D (PCNA staining) denotes the relative change in PCNA level in 24 and 48 hours of etoposide treatment with respect to DMSO treated cells. We could find that PCNA fluorescence was higher after 24 hours of etoposide treatment which reduced slightly in the 48 hours timepoint. This reduction has been analysed and found to be statistically significant. The data shown here is based on the experiment to optimise the duration of etoposide treatment. Hence, all the experiments afterwards have been performed considering 24 hours of etoposide treatment. We have inserted this information in line no 329-331 in the revised manuscript.

  • The list of differentially expressed genes and corresponding pathways (422-456) would be easier to assimilate in a table format.

Response: We thank the reviewer for this comment. We have assimilated in a table format as suggested.

  • In Suppl information file 1, the list of antibodies should also include catalog numbers along with the company it was purchased from.

Response: We thank the reviewer for this comment. We have mentioned the catalog numbers as suggested.

  • While detecting more S-phase nuclei in Lamin A downregulated OVCAR3, did the authors look at if PCNA levels were altered? They already had done this for etoposide treated OVCAR3, Figure 3D and it would support their claim if PCNA levels altered in lamin1 knockdown.

Response: We thank the reviewer for this comment. We have checked the levels of PCNA after treatment in LMNA knockdown OVCAR3 cells. The figure has been inserted in supplementary figure 6.

Round 2

Reviewer 1 Report

It is incorrect the statement in the Abstract that “alteration in lamin A/C expression and distribution drives tumorigenesis of almost all tissues of human bodies”.

The fact that OVCAR3 have higher expression of lamin A vs IOSE does not mean that lamin A is responsible for DNA repair alterations. Many other proteins are differentially expressed in the two cell lines, as it is shown in the RNAseq analysis, which can be responsible for DNA repair defects. Unless the authors check whether knockdown of lamin A reverts the DNA repair phenotype, no conclusions can be drawn. 

Why are BRCA1 and BRCA2 localized at the cytoplasm in some of the IF figures? Look at Figure 2A. Also, in Figure 2B, are they quantifying DNA repair proteins in the whole cell or only in the nucleus? How many cells were counted? Moreover, in Figure 2C, it is not clear which protein is Rad51 in the Western blot. It seems that they did two antibodies at once, or that they reblotted.  

What is the point of Figure 3? Etoposide treatment in all cells causes DNA damage and in OVCAR cells does too. This should be placed in supplemental data as control experiments.

Authors perform RNAseq in OVCAR3 cells treated with etoposide or vehicle control and differential gene expression is presented. As expected, many pathways linked to DNA repair and genome stability are identified. Also, genes in the PI3K-AKT signaling pathway are identified. I am not sure what advancement these data provide.

The key question of the paper -the role of increased lamin A levels- is addressed in Figure 6, as they test the impact of lamin A knockdown. Reported that depletion of lamin A leads to decreased levels of all transcripts levels tested, expect for one, PLK1. However, this reviewer finds quality control issues with this figure. First, figure 6C, viability assay shows that untreated cells depleted of lamin A are already less viable and with etoposide the difference in viability is minimal. Also, in figure 6E, not clear why the fold change in gene expression is 1 in scrambled sham control and in siLMNA sham control. 

Overall, the study seems a bit preliminary for publication in Cells. 

Reviewer 2 Report

  1. Line 276-308 “Increased lamin A expression in high-grade ovarian carcinoma cell lines (OVCAR3): 

a.     The authors should revise Figures 1a and b. The representative figure 1a should be a group of three to four cells, not a single-cell image. Also, the image look stretched so my advice is to avoid the extra stretch. What dpi authors have chosen? 

b.     Figure 1b, loading control is not accurate and representative blots have a lot of background noise. Authors need to provide good and clear representative blots and I would like to see the full blot in supplementary figures.

Although, authors improved the clarity of the blot but still the representative blot suggest me the unequal loading control and also the blot is cropped very tightly. Also, Figure 1B and C and others axis are not clear. I would suggest the authors to use Graph Pad for statistical analysis  as I mentioned in my previous report.

2.     Line 309-336; authors observed no change in Ku70 expression. I am wondering if the authors checked the ATM and ATR expression also. It would be more supportive if authors can check their expression as Bradburry et al 2020 also observed the low expression of ATM in OVCAR3 cells. 

Response: We thank the reviewer for this comment. The level of Ku70 remains the same in IOSE and OVCAR3 which was also established earlier by Langland et al, 2010. We have added the reference in line no 299. As, Ku70 level does not change between OVCAR3 and IOSE, We have not checked the expressions of ATM and ATR as reported earlier .

I am not convince with authors response as there are evidences available which demonstrated that Lamin A/C also plays a role in DNA damage, repair, and apoptosis. and show increased double-strand break marker γ-H2AX, and defective localization of DNA damage regulators ATR and ATM and repair factors Rad 50 and Rad 51. 

3.     I am wondering if the authors checked the expression of lamin C also.

Response: We thank the reviewer for this comment. The antibody used in this study for lamin A is Anti-Lamin A (C-terminal) antibody produced in rabbit (L1293)(Sigma -Aldrich) which shows only one band for lamin A in 74kD. So, we have not checked the expression of lamin C. 

I would strongly suggest to authors to check the Lamin C expression also as there are reports suggest that although  Lamin A and Lamin C levels are equal in cells generally , differences have been found in some different cancer tissues.

4.      I am also confused about the lamin A staining in Figure 2a. Images suggest that lamin A expression is low in OVCAR3 cells compared to IOSE cells and this expression profile is totally reversed as authors shown in Figure 1a. 

Response: We thank the reviewer for this comment. We clarify that we had distinctly shown an elevated expression of lamin A in OVCAR3 compared to IOSE. We quantified the same using ImageJ mentioned in the method section in details. We would like to draw the attention of the reviewer to figure 2a where the top panel exhibiting lower expression lamin A corresponds to IOSE and the bottom panel showing elevated expression of lamin A corresponds to OVCAR3. Hence, there is nothing to revise in this figure.

I am still confuse about the Lamin A expression. In figure 1a authors have shown the higher expression of lamin A in OVCAR3 cell lines compared to IOSE. However, in figure 2a Images suggest that lamin A expression is low in OVCAR3 cells compared to IOSE cells. Also, authors need to improve the RAD51 blot as the representative blot is  not clear.

5.     I am not convinced with the image quality and it should be more clear. In Figure 3d. as per my understanding, the representative Image of PCNA suggests the high expression at 48 hr. however, the authors' statistical data is the reverse. The authors have to check their analysis again.

Response: We thank the reviewer for this comment. The bar diagram in figure 3D (PCNA staining) denotes the relative change in PCNA level in 24 and 48 hours of etoposide treatment with respect to DMSO treated cells. We could find that PCNA fluorescence was higher after 24 hours of etoposide treatment which reduced slightly in the 48 hours timepoint. This reduction has been analysed and found to be statistically significant. This experiment was performed initially to optimise the duration of etoposide treatment. All the experiments afterward have been performed considering 24 hours  of etoposide treatment

I am still not convinced. I am wondering how author’s quantified the  PCNA foci?

17. I struggle with the data presented in Figures 6b and 6c. The authors have shown that the etoposide treatment in knockdown cells did not change the BrdU expression levels, However, based on the image it gives me the impression that BrdU expression is markedly induced in knockdown cells compared with non-treated cells. The authors need to explain this. 

Response: We thank the reviewer for this comment. The image in Figure 6B shows a prominent drop in the number of BrdU positive cells in LMNA knock down OVCAR3 cells and the difference in the number of BrdU-positive cells was insignificant between the etoposide-treated and untreated counterparts of LMNA-deficient OVCAR3 which indicates that first of all etoposide triggers S phase accumulation (Nam, C., K. Doi, and H. Nakayama, Etoposide induces G2/M arrest and apoptosis in neural progenitor cells via DNA damage and an ATM/p53-related pathway. Histol Histopathol, 2010. 25(4): p. 485-93) and secondly lamin A drives the process positively. This observation has been revalidated by PCNA staining in etoposide treated OVCAR3 cells before and after knockdown. PCNA fluorescence was much lower in etoposide treated LMNA knock down OVCAR3 cells (Supplementary Figure 6).

I am still not convinced by the authors response. Representative images still gives me the impression that BrdU expression is markedly induced in knockdown cells compared with non-treated cells.

Reviewer 3 Report

The authors have addressed most concerns, thus benefiting the manuscript. For the major points- taking into consideration that the validation by co-IP cannot be performed, at least some conclusion lines the revised text discussion such as ‘that the study for the first time provides  a molecular orchestration’, is a strong  statement that should be edited, to make it adhere more to what was done. The qPCR is sufficient to state that the expression of those genes are affected, but it does not really support that they are ‘molecular interactors’.

Some minor points-

It is highly encouraged for authors to use color schemes that are useful for colorblind readers, and not adhere to the green/red scheme.

Line 313- The phrase the experiments were repeated multiple times is not a concrete scientific statement. The number of cells used, number of biological replicates done should be clearly written in the methods/legends and does not have to be repeated in the results section.

Line 325- correct  to “…from each is selected as representative…”
